# The calcineurin pathway regulates extreme thermotolerance, cell membrane and wall integrity, antifungal resistance, and virulence in *Candida auris*

**Hyunjin Cha[1], Doyeon Won[1], Seun Kang[2], Eui-Seong Kim[2], Kyung-Ah Lee[3,4], Won-Jae Lee[3], Kyung-Tae Lee[2]\*, Yong-Sun Bahn[1]\***

1 Department of Biotechnology, College of Life Science and Biotechnology, Yonsei University, Seoul, Republic of Korea, 2 Korea Zoonosis Research Institute, Jeonbuk National University, Iksan, Jeonbuk, Republic of Korea, 3 School of Biological Sciences, Seoul National University, Seoul, South Korea, 4 Saeloun Bio Inc., Seoul, South Korea

\* lee.kt@jbnu.ac.kr (K-TL); ysbahn@yonsei.ac.kr (Y-SB)

## Abstract

*Candida auris*, an emerging fungal pathogen characterized by its multidrug resistance and high mortality rates, poses a significant public health challenge. Despite its importance, the signaling pathways governing virulence and antifungal resistance in *C. auris* remain poorly understood. This study investigates the calcineurin pathway in *C. auris*, critical for virulence and antifungal resistance in other fungal pathogens. Calcineurin, a calcium/calmodulin-dependent protein phosphatase, comprises a catalytic subunit (Cna1) and a regulatory subunit (Cnb1) in *C. auris*. Our findings reveal that deletion of *CNA1* or *CNB1* disrupts extreme thermotolerance and cell membrane and wall integrity, leading to increased susceptibility to azoles and echinocandins. Moreover, we identified a downstream transcription factor, Crz1, which plays a central role in this pathway in other fungal species. Deletion of *CRZ1* resulted in thermotolerance and membrane integrity defects comparable to those of *cna1Δ* and *cnb1Δ* mutants, along with increased azole susceptibility. Supporting it, fluconazole treatment induced Crz1 nuclear translocation in a Cna1-dependent manner. However, unlike *cna1Δ* and *cnb1Δ* mutants, the *crz1Δ* mutant displayed increased resistance to echinocandins, suggesting the opposing roles for Crz1 in regulating cell wall integrity. Nevertheless, echinocandins also promoted Crz1 nuclear translocation via Cna1, underscoring the complex regulatory mechanisms at play. Cna1 was found to be required for virulence in both the *Drosophila* systemic infection model and the murine skin infection model. However, in a systemic murine infection model, both calcineurin and Crz1 appeared dispensable for *C. auris* virulence. Our findings highlight that the evolutionarily conserved calcineurin pathway employs distinct regulatory mechanisms to perform divergent roles in regulating extreme thermotolerance, cell wall and membrane integrity, antifungal drug resistance, and virulence in *C. auris*.

**Data availability statement:** All data are available in the manuscript and the Supporting Information files. Other supporting raw data files were uploaded to figshare as follows: https://figshare.com/s/0aa6a39d8b-7b6334a936. DOI: dx.doi.org/10.6084/m9.figshare.28148153.

**Funding:** This work was supported by the National Research Foundation of Korea (NRF) grant funded by the Korean government (MSIT) (RS-2021-NR058017, RS-2021-NR056582, RS-2025-00555365, and RS-2025-02215093 to YSB; RS-2022-NR072215 and RS-2021-NF000550 to KTL; RS-2024-00345184 to WJL) and by the Yonsei Signature Research Cluster Program (2025-22-0015 to YSB). The funders had no role in study design, data collection and analysis, decision to publish, or preparation of the manuscript.

**Competing interests:** The authors have declared that no competing interests exist.

## Author summary

The fungal pathogen *Candida auris* presents a global health threat due to its multidrug resistance and high mortality rates. Despite its clinical significance, the molecular mechanisms underlying its virulence and antifungal resistance remain poorly understood. This study investigates the complex role of the calcineurin signaling pathway in *C. auris* pathogenicity. Deletion of the calcineurin complex impairs extreme thermotolerance and compromises cell membrane and wall integrity, leading to increased susceptibility to azoles and echinocandins, antifungal agents targeting the cell membrane and wall, respectively. We also identified the Crz1 transcription factor as a downstream target of calcineurin signaling. Interestingly, unlike calcineurin mutants, Crz1 mutants are susceptible only to cell-membrane-targeting azoles but surprisingly exhibit increased resistance to cell-wall-targeting echinocandins, suggesting that the calcineurin pathway may regulate multiple transcription factors, including Crz1. Additionally, calcineurin was found to be essential for virulence *in vivo*, using two different animal infection models, fruit fly and mice. These findings highlight the essential role of the calcineurin pathway in *C. auris* virulence, offering novel insights into its role in antifungal resistance and virulence, and paving the way for targeted therapies.

## Introduction

Fungal infections pose a significant global health threat, accounting for an estimated 3.75 million deaths annually, with nearly 1 million of these attributable to *Candida* bloodstream infections or invasive candidiasis [1]. The fungal pathogen *Candida auris* has emerged as a major clinical concern due to its ability to infect immunocompromised patients and its rapid spread in healthcare settings, with reported mortality rates approaching 34% [2]. *C. auris* often shows resistance to several antifungal drugs, including azoles, amphotericin B, and echinocandins, which are commonly used for candidiasis treatment. In 2020, the CDC reported that 86% of *C. auris* isolates in the United States are resistant to azoles, 26% to amphotericin B, and about 1% to echinocandins [3]. Due to these factors, the World Health Organization (WHO) has classified *C. auris* as a critical priority fungal pathogen. Consequently, understanding its mechanisms of virulence and antifungal resistance is crucial for developing new antifungal therapies.

The calcineurin pathway is an evolutionarily conserved eukaryotic signaling cascade. Calcineurin is a serine/threonine protein phosphatase composed of catalytic (Cna1) and regulatory (Cnb1) subunits. It is activated when the calcium ions ($Ca^{2+}$) enter the cytosol through membrane transporters. The influx of $Ca^{2+}$ is sensed by calmodulin, leading to the formation of the $Ca^{2+}$-calmodulin complex that binds to Cna1. Concurrently, cytosolic $Ca^{2+}$ also binds to Cnb1, inducing a conformational change in Cna1 that facilitates its interaction with the $Ca^{2+}$/calmodulin complex, thereby activating calcineurin [4]. Following activation, calcineurin dephosphorylates

a downstream transcriptional factor. This factor then translocates from the cytosol to the nucleus, regulating the expression of calcineurin-dependent genes [4]. In humans, calcineurin targets NFAT (nuclear factor of activated T cells), while in yeast, the target is Crz1 (calcineurin-responsive zinc finger 1) [5].

The calcineurin-Crz1 pathway plays a critical role in environmental adaptation in yeast and various fungal pathogens. In *Saccharomyces cerevisiae*, calcineurin is indispensable for responding to high concentrations of cations [6]. Furthermore, the deletion of *CRZ1* results in phenotypes similar to those of calcineurin mutants, though less severe [7]. In *Candida albicans*, the calcineurin pathway is crucial for regulating cell wall integrity, serum survival, virulence, and drug tolerance [4,8,9]. Similarly, in *Candida glabrata*, it influences thermotolerance, intracellular architecture, and virulence [10]. In *Cryptococcus neoformans*, the calcineurin pathway is essential for stress adaptation and virulence, particularly under high-temperature conditions, impacting cell wall remodeling, calcium transport, and pheromone production [6,11]. Additionally, in *C. neoformans*, substrates other than Crz1 have been identified as targets of calcineurin [12].

Despite the conserved and divergent roles of the calcineurin pathway in the growth, development, antifungal drug resistance, and pathogenicity across other fungal pathogens, its pathobiological functions and regulatory mechanism remain unexplored in *C. auris*. In this study, we identified and functionally characterized the catalytic and regulatory subunits of calcineurin in *C. auris*. Here, we demonstrate that the calcineurin pathway is required for governing extreme thermotolerance, cell membrane and wall integrity, resistance to azoles and echinocandins, and virulence in *C. auris*. We also discovered that Crz1 is one of the calcineurin targets but plays overlapping and distinct roles with calcineurin. Collectively, our study highlights that the calcineurin pathway plays conserved and distinct pathobiological roles with distinct regulatory mechanisms in this emerging multidrug-resistant fungal pathogen.

## Results

### Calcineurin is essential for extreme thermotolerance and the maintenance of cell membrane and wall integrity in *C. auris*

To identify the catalytic (Cna1) and regulatory (Cnb1) subunits of calcineurin in *C. auris*, we queried the *Candida* genome database (http://www.candidagenome.org/). Orthologs were identified in the clade I *C. auris* wild-type strain (AR0387) as Cna1 (B9J08_004950) and Cnb1 (B9J08_001732). Sequence analysis revealed that both subunits are highly conserved across eukaryotic species (S1A and S1B Fig). Protein domain analysis using the InterPro database (https://www.ebi.ac.uk/interpro/) showed that Cna1 harbors a conserved serine/threonine-specific protein phosphatase domain, which catalyzes the dephosphorylation of phosphoserine and phosphothreonine residues. The size of Cna1 is comparable to its orthologs in other fungal species (Fig 1A). Cnb1 was found to contain four EF-hand motifs, characteristic calcium-binding domains (Fig 1A). Structural modeling using AlphaFold3 predicted a heterodimeric calcineurin complex in *C. auris* consisting of Cna1 and Cnb1, featuring helix-loop-helix structures (Fig 1B) [13,14].

To investigate the roles of calcineurin in *C. auris*, we generated *cna1Δ*, *cnb1Δ*, and *cnb1Δ cna1Δ* mutants in the clade I AR0387 (B8441) strain background (S2A, S2B, and S2C Fig). To confirm the phenotypic features of the mutants, we also constructed complemented strains by integrating the wild-type allele into its original locus (*cna1Δ::CNA1* and *cnb1Δ::CNB1*) (S2D and S2E Fig). Under nutrient-rich growth conditions at ambient temperatures (25°C to 30°C), the calcineurin mutants exhibited no growth defects (Fig 1C and 1D), indicating that the calcineurin pathway is not essential for the normal growth of *C. auris*. As the calcineurin pathway has been implicated in thermotolerance, we assessed the growth of the mutants at elevated temperatures (37°C to 42°C) [12]. The *cna1Δ*, *cnb1Δ*, and *cnb1Δ cna1Δ* mutants showed no growth defects at high temperatures within the physiological range (37–39°C) (Fig 1C and 1D), suggesting that the calcineurin pathway is largely dispensable for thermotolerance of *C. auris* under these conditions. However, under severe heat stress conditions exceeding 42°C (43–45°C), *cna1Δ*, *cnb1Δ*, and *cnb1Δ cna1Δ* mutants exhibited significant growth defects (Fig 1E and 1F), indicating the critical role of calcineurin in regulating growth at extreme temperatures.

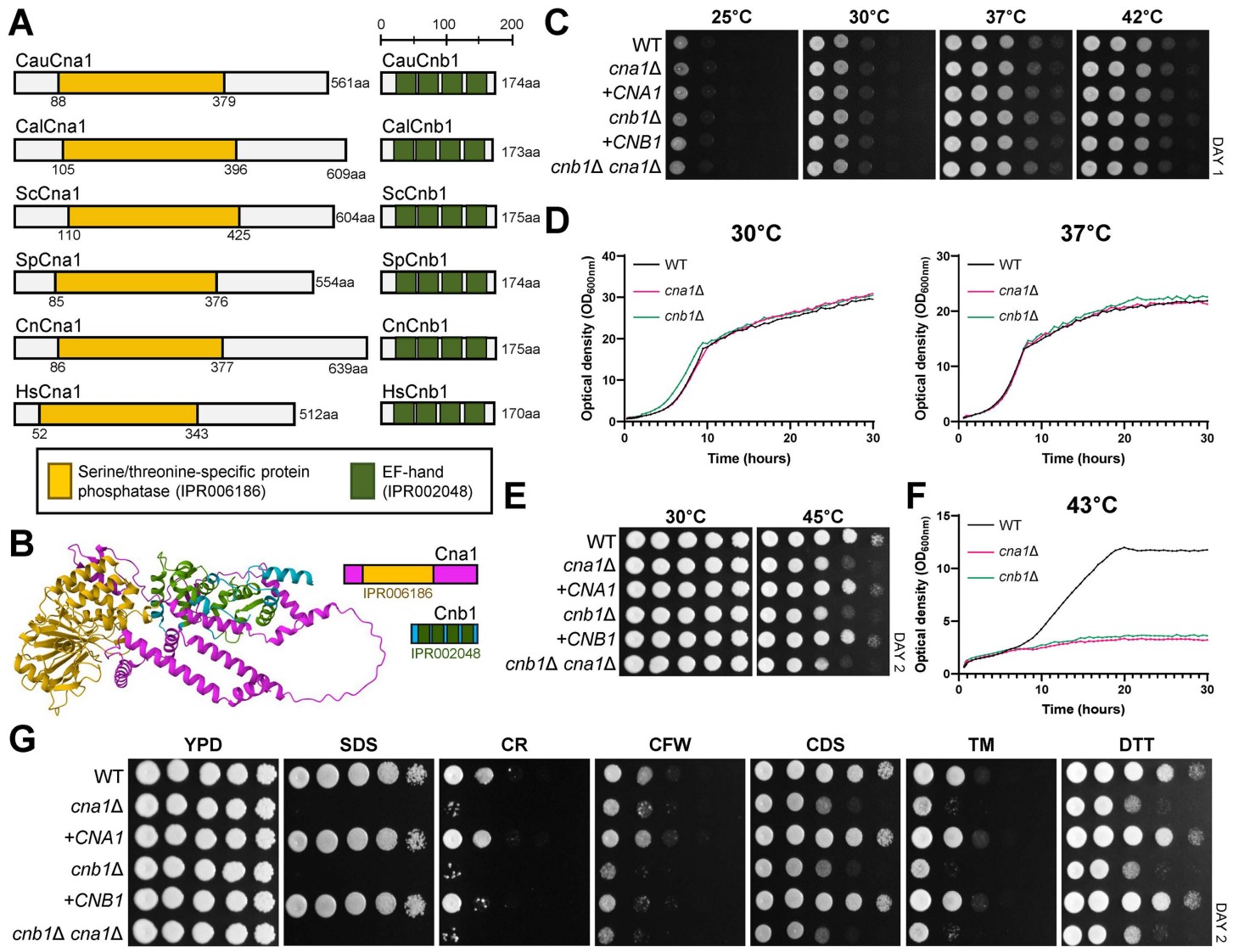

**Fig 1. Identification of the calcineurin complex and its roles in growth, thermotolerance, and stress responses in *C. auris*.** (A) Domain analysis of Cna1 and Cnb1 in *Candida auris* (Cau), *Candida albicans* (Cal), *Saccharomyces cerevisiae* (Sc), *Schizosaccharomyces pombe* (Sp), *Cryptococcus neoformans* (Cn), and *Homo sapiens* (Hs) was performed using InterPro (https://www.ebi.ac.uk/interpro/). (B) Predicted structures of *C. auris* Cna1 and Cnb1 were modeled using AlphaFold 3. The protein sequences were processed in the order of Cna1 and Cnb1. The color coding is as follows: Cna1 in pink, the IPR006186 domain of Cna1 in yellow, Cnb1 in blue, and the IPR02048 domain of Cnb1 in green. (C) Qualitative spot assays display the growth of the following strains under varying temperatures: wild-type (WT; B8441), *cna1Δ* (YSBA99), *cna1Δ::CNA1* (+*CNA1*; YSBA110), *cnb1Δ* (YSBA102), *cnb1Δ::CNB1* (+*CNB1*; YSBA111), and *cnb1Δ cna1Δ* (YSBA172). WT and mutant strains were cultured overnight in liquid YPD medium at 30°C, serially diluted 10-fold, and spotted on YPD agar plates. Plates were incubated for 1 day at 25°C, 30°C, 37°C, or 42°C. (D) Quantitative growth rates of the WT, *cna1Δ*, and *cnb1Δ* strains were measured at 30°C and 37°C using a multichannel bioreactor (Biosan Laboratories, Inc., Warren, MI) by measuring the $OD_{600}$ for 30 h. (E) WT and mutant strains were spotted on YPD plates and incubated at 30°C or 45°C for 2 days. (F) Growth rates of the WT and mutant strains were monitored at 43°C. (G) WT and mutant strains were spotted on YPD medium supplemented with stress-inducing agents, including 0.1% sodium dodecyl sulfate (SDS), 0.05% Congo red (CR), 3 mg/mL calcofluor white (CFW), 250 mM $CdSO_4$, 2.5 μg/mL tunicamycin (TM), or 22 mM dithiothreitol (DTT). Plates were incubated for 2 days at 30°C.

We next investigated whether the calcineurin pathway is required for the growth of *C. auris* under various environmental stress conditions, including osmotic stress, oxidative stress, genotoxic stress, and cell membrane/wall-damaging stress (Figs 1G and S3). Among these stressors, the *cna1Δ*, *cnb1Δ*, and *cnb1Δ cna1Δ* mutants demonstrated markedly enhanced sensitivity to cell membrane-destabilizing agents [sodium dodecyl sulfate (SDS)] and cell wall damaging agents [dithiothreitol (DTT), tunicamycin (TM), Congo red (CR), calcofluor white (CFW), and CdSO$_4$ (CDS)] (Fig 1G). SDS induces cell membrane stress by disrupting lipid bilayers [15], while CR binds specifically to polysaccharide components such as chitin and glucans in the fungal cell wall [16]. DTT and TM, as ER stressors, cause protein misfolding, leading to compromised cell wall integrity [17]. CdSO$_4$, a toxic heavy metal, interferes with the synthesis and structure of fungal cell wall components such as chitin and glucans, thereby disrupting cell wall integrity [18]. Notably, deletion of both *CNA1* and *CNB1* resulted in phenotypes identical to those observed with individual deletions of *CNA1* or *CNB1*, suggesting that Cna1 and Cnb1 function as components of a single calcineurin complex, as expected. In summary, our findings demonstrate that the calcineurin pathway plays a crucial role in extreme thermotolerance and the maintenance of the cell membrane and wall integrity in *C. auris*.

## Calcineurin promotes tolerance to azoles and echinocandins in *C. auris*

Given the role of calcineurin in maintaining cell membrane and wall integrity, we investigated its contribution to tolerance against clinically relevant antifungal drugs, which target either the fungal cell membrane (azoles and polyenes) or cell wall (echinocandins). We evaluated the susceptibility of the mutants (*cna1Δ*, *cnb1Δ*, and *cnb1Δ cna1Δ*) and their complemented strains to echinocandins [caspofungin (CAF), micafungin (MIF), and anidulafungin (ANF)], polyenes [amphotericin B (AMB)], and azoles [fluconazole (FLC), posaconazole (PSC), voriconazole (VRC), ketoconazole (KTC), and itraconazole (ITC)] (Fig 2A). The *cna1Δ*, *cnb1Δ*, and *cnb1Δ cna1Δ* mutants exhibited substantially increased susceptibility to all tested echinocandins and azoles compared to the wild-type strain (Fig 2A). The minimum inhibitory concentrations (MICs) of CAF, MIF, and ANF for the wild-type strain were > 80 µg/mL, 1.25 µg/mL, and 1.25 µg/mL, respectively (Figs 2B and S4A). In contrast, the MICs for calcineurin mutants were markedly reduced: 5 µg/mL for CAF, 0.625 µg/mL for MIF, and 0.3125 µg/mL for ANF (Figs 2B and S4A). Consistent with previous reports [19], azoles exhibited fungistatic activity against wild-type *C. auris* and did not completely inhibit its growth (Fig 2B). However, in the absence of calcineurin (*cna1Δ*, *cnb1Δ*, and *cnb1Δ cna1Δ*), azoles displayed fungicidal activity (Fig 2B).

The observation that the calcineurin pathway promotes tolerance to azoles and echinocandins in *C. auris* suggests that calcineurin inhibitors, such as FK506 (tacrolimus) and cyclosporin A (CsA), may exhibit synergistic antifungal activity when combined with these drugs. To test this hypothesis, we performed checkerboard assays with FK506, CsA, azoles, and echinocandins. As anticipated, given our findings that deletion of *CNA1* and *CNB1* does not impair the growth of *C. auris* under basal conditions, FK506 and CsA alone displayed no antifungal activity at concentrations up to 16 and 24 µg/mL, respectively (Figs 2C and S4B). However, FK506 and CsA exhibited fungicidal activity only when combined with echinocandins, suggesting a synergistic effect. The drug combinations were tested using the following echinocandin concentrations: CAF (up to 20 µg/mL), MIF (1 µg/mL), and ANF (1 µg/mL). Assuming an MIC of 16 µg/mL for FK506 and 24 µg/mL for CsA, the fractional inhibitory concentration (FIC) indices for their combinations with echinocandins were as follows: FK506 combined with CAF, MIF, and ANF yielded FIC indices of <0.03, 0.14, and <0.08, respectively. Similarly, CsA combined with CAF, MIF, and ANF yielded FIC indices of <0.13, 0.14, and <0.1. All FIC indices were consistently below 0.5 (Figs 2C, S4B and S4C), indicating strong synergistic interactions. Additionally, FK506 and CsA exhibited synergistic fungicidal antifungal activity when combined with azoles. The FIC indices were <0.08 (FK506 + FLC), <0.08 (CsA + FLC), <0.03 (FK506 + PSC), <0.04 (CsA + PSC), <0.15 (FK506 + VRC), <0.16 (CsA + VRC), 0.05 (FK506 + ITC), and 0.06 (CsA + ITC) (Figs 2D and S5).

Unexpectedly, we found that the *cna1Δ*, *cnb1Δ*, and *cnb1Δ cna1Δ* mutants exhibited high resistance to AMB, in stark contrast to their heightened susceptibility to azoles (Fig 2A). Azoles inhibit ergosterol synthesis by targeting lanosterol

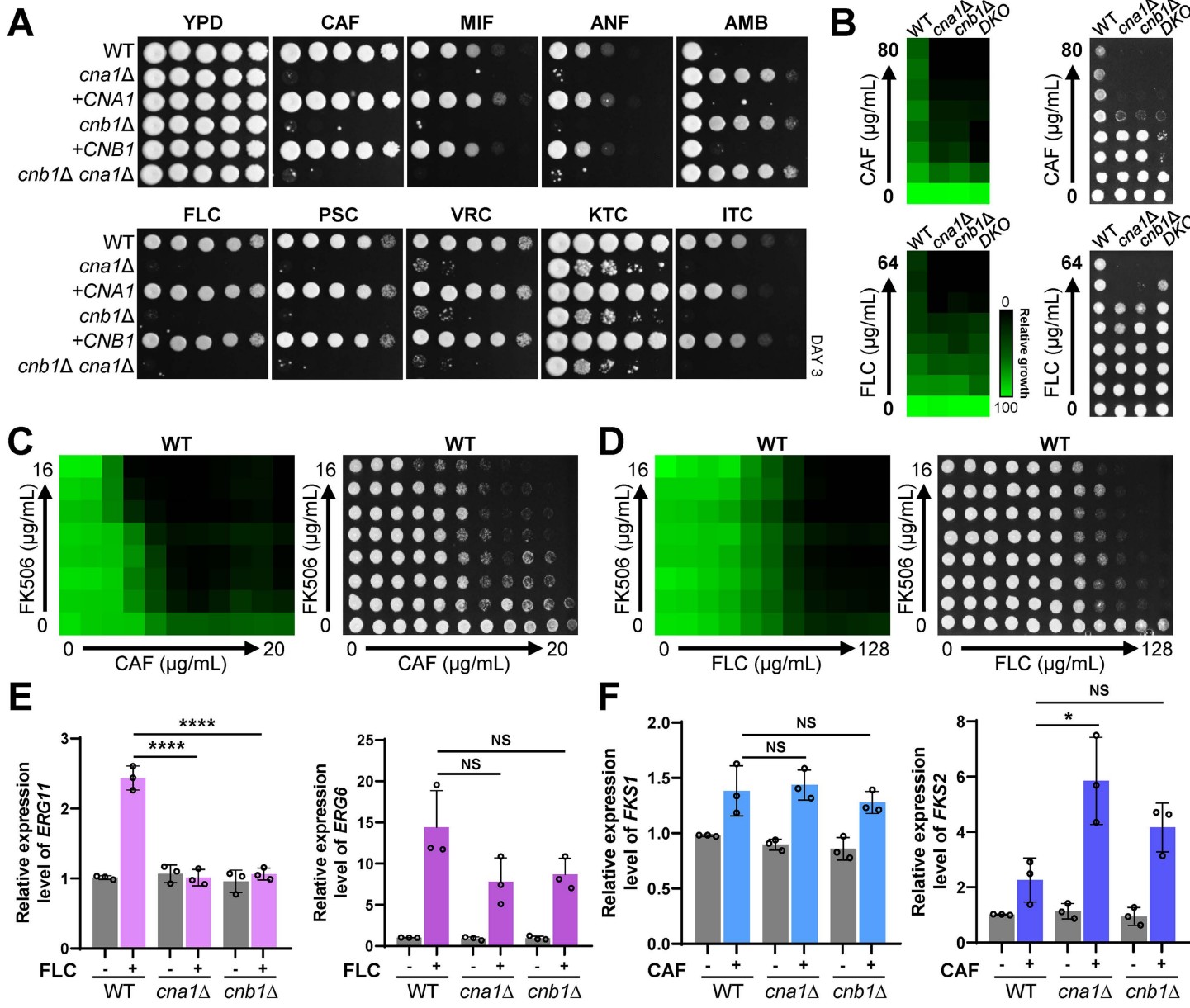

**Fig 2. Functions of calcineurin in antifungal tolerance in *C. auris*.** (A) Qualitative spot assays showing the stress susceptibility of the following strains: WT (B8441), *cna1Δ* (YSBA99), +*CNA1* (YSBA110), *cnb1Δ* (YSBA102), +*CNB1* (YSBA111), and *cnb1Δ cna1Δ* (YSBA172). WT and mutant strains were cultured overnight in liquid YPD medium at 30°C, serially diluted 10-fold, and spotted on YPD agar medium supplemented with stressors, including 2.5 µg/mL caspofungin (CAF), 0.15 µg/mL micafungin (MIF), 1 µg/mL anidulafungin (ANF), 3 µg/mL amphotericin B (AMB), 150 µg/mL fluconazole (FLC), 0.5 µg/mL posaconazole (PSC), 0.8 µg/mL voriconazole (VRC), 5 µg/mL ketoconazole (KTC), or 1 µg/mL itraconazole (ITC). Plates were incubated at 30°C for 3 days. (B) EUCAST MIC test results of CAF and FLC in the WT, *cna1Δ*, *cnb1Δ*, and *cnb1Δ cna1Δ* (referred to as *DKO*) strains. The strains were grown in YPD medium overnight, washed twice with $H_2O$, resuspended in $H_2O$, and added (100 µL) to 10 mL of liquid RPMI 1640 medium, which was subsequently loaded into 96-well plates. Drugs serially diluted 2-fold from the indicated concentrations were added to each well. To evaluate fungicidal activity, cultures were spotted onto YPD agar plates, followed by a 24 h incubation at 30°C. (C) Checkerboard assay results of CAF with FK506 in the WT strain. The assay was performed using serial two-fold dilutions of CAF, with concentrations ranging from 0 to 20 µg/mL. After growth measurement, cultures were spotted onto YPD agar plates and incubated at 30°C for 24 h to evaluate fungicidal activity. An FIC index of <0.03 was observed. (D) Checkerboard assay results of FLC with FK506 in the WT strain. The FIC index was determined to be <0.08. (E) qRT-PCR analysis of *ERG11* and *ERG6* in the WT, *cna1Δ*, and *cnb1Δ* strains. The strains were cultured for 24 h in the presence of FLC (200 µg/mL) at 30°C in a shaking incubator. (F) qRT-PCR analysis of *FKS1* and *FKS2* in the WT, *cna1Δ*, and *cnb1Δ* strains. The strains were cultured overnight at 30°C shaking incubator,

sub-cultured to OD$_{600}$ 0.8 in fresh YPD liquid medium, and then cultured for 24 h with CAF (5 μg/mL) at 30°C shaking incubator. Gene expression levels were normalized to *ACT1*, and fold changes were calculated relative to the basal expression level in WT strain. Statistical significance was determined using one-way ANOVA with Bonferroni's multiple-comparison test (*, $P < 0.05$; ****, $P < 0.0001$; NS, not significant).

14α-demethylase (Erg11), while AMB binds to and sequesters membrane ergosterol. A possible explanation for this finding is that the calcineurin pathway may regulate *ERG11* expression in *C. auris*, as reduced membrane ergosterol could confer resistance to AMB. To explore this hypothesis, we assessed basal *ERG11* expression levels, which were comparable between the wild-type and calcineurin mutant strains (Fig 2E). In response to FLC treatment, *ERG11* expression was upregulated in the wild type, possibly as a compensatory mechanism. However, this induction was absent in the *cna1Δ* and *cnb1Δ* mutants (Fig 2E), suggesting that the calcineurin pathway plays a role in the transcriptional regulation of ergosterol biosynthetic genes associated with FLC tolerance. By contrast, *ERG6* expression was induced in the calcineurin mutants upon FLC treatment, suggesting that not all ergosterol biosynthetic genes are regulated by the calcineurin pathway. In contrast, there were no appreciable changes in basal expression of *ERG* genes or in total ergosterol content in the calcineurin mutant strains (Figs 2E and S6). Moreover, even after AMB treatment, total ergosterol levels in calcineurin-deficient strains remained comparable to those in the wild-type strain (S6 Fig), indicating that altered ergosterol abundance is unlikely to account for the AMB resistance observed.

Interestingly, membrane permeability, which could influence azole susceptibility, was slightly increased in the calcineurin mutants, potentially promoting greater intracellular accumulation of FLC (S7 Fig). In contrast, AMB resistance may instead be attributed to changes in cell wall and membrane structure rather than ergosterol levels. These findings support the conclusion that differential responses to FLC and AMB in the calcineurin mutants are likely driven by distinct cellular mechanisms, including altered membrane permeability and cell wall architecture.

We further investigated echinocandin tolerance, which is associated with altered expression of *FKS1* and *FKS2*, encoding β-1,3-glucan synthase, the target of echinocandins. Basal and echinocandin-induced expression levels of *FKS1* in the wild type, *cna1Δ*, and *cnb1Δ* mutants, revealing no significant changes in *FKS1* expression (Fig 2F). In contrast, *FKS2* expression was markedly upregulated in the *cna1Δ* mutant upon echinocandin treatment (Fig 2F), probably as a compensatory response aimed at restoring cell wall integrity in the absence of calcineurin function (Fig 2F). The increased susceptibility of the *cna1Δ* mutant to echinocandins is more likely attributable to an inherent defect in maintaining cell wall integrity resulting from the loss of calcineurin function. Collectively, our findings demonstrated that the calcineurin pathway is critical for promoting tolerance to azoles and echinocandins in *C. auris*.

## Conserved roles of calcineurin in other *C. auris* clades

*Candida auris* is classified into five clades according to the isolated location: clade I (South Asia), clade II (East Asia), clade III (Africa), clade IV (South America), and clade V (Iran) [20]. Each clade exhibits distinct genotypic and phenotypic variations [21,22]. To determine whether the role of the calcineurin pathway is conserved among different clades of *C. auris*, we performed a phylogenetic analysis, which revealed that Cna1 is highly conserved across all clades (Fig 3A).

To validate the functions of Cna1 in clades other than clade I, we constructed *CNA1* knockout mutants in strains B11220 (clade II), B11221 (clade III), and B11245 (clade IV) (S8 Fig). Spot assays were conducted to evaluate the mutants' responses to various stress conditions. The stress response phenotypes of these mutants were largely consistent with those of B8441 (clade I), except for differences observed in their sensitivity to the cell wall-destabilizing agent, CFW (Fig 3B and 3C). Unlike *cna1Δ* mutants in clades I and II, *cna1Δ* mutants in clades III and IV exhibited increased resistance to CFW, implying that the calcineurin pathway may have clade-specific roles in maintaining cell wall integrity in *C. auris*.

To investigate whether the phenotypic variation observed among *cna1Δ* mutants could be attributed to differences in *CNA1* expression, we measured both basal and CFW-induced *CNA1* transcript levels in wild-type strains from each clade.

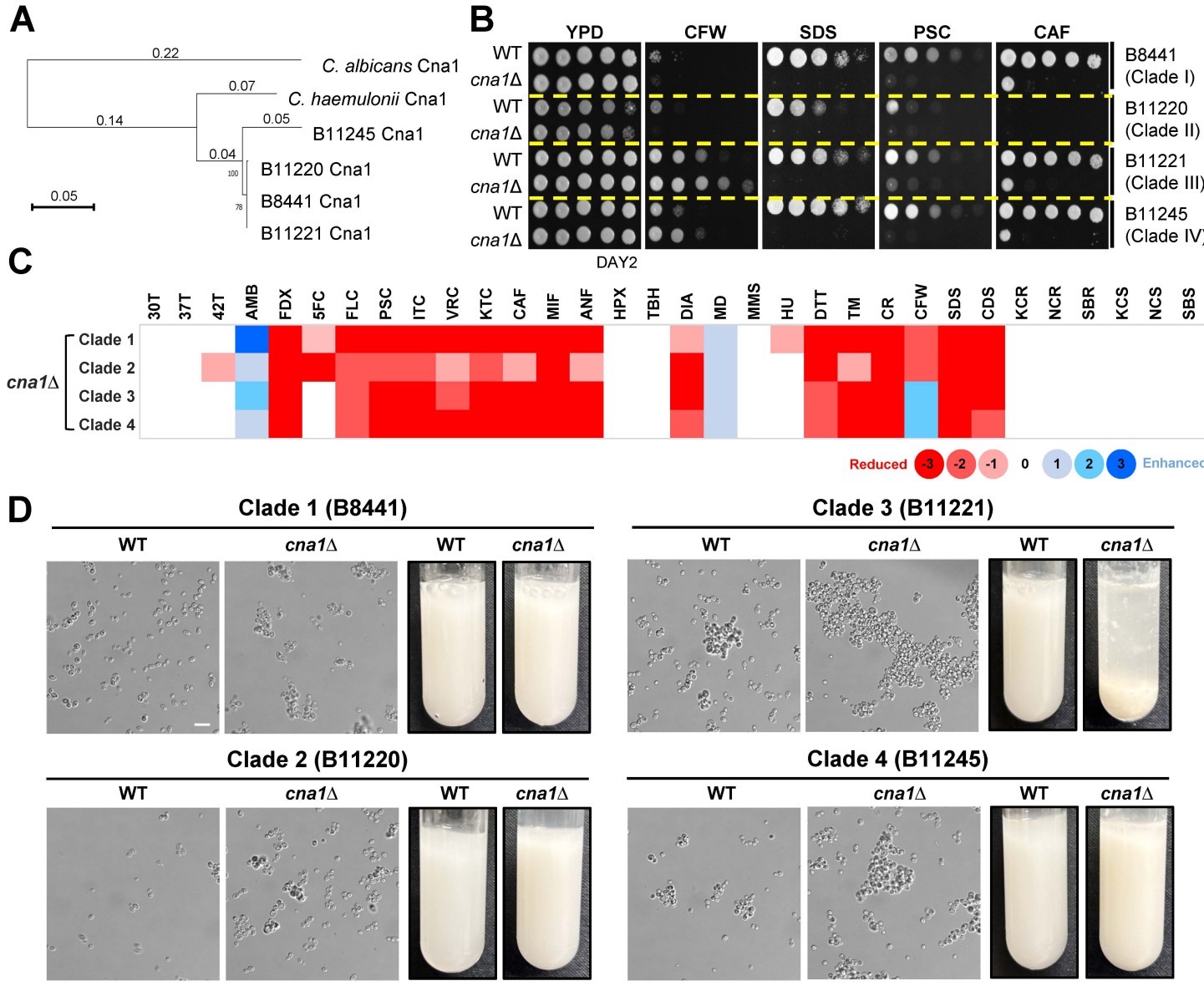

**Fig 3. Functional roles of calcineurin across *C. auris* clades.** (A) Phylogenetic analysis of Cna1 orthologs across *C. albicans*, *C. haemulonii*, and four *C. auris* clades. (B) Qualitative spot assays showing the stress susceptibility of clade I WT (B8441), clade I *cna1Δ* (YSBA99), clade II WT (B11220), clade II *cna1Δ* (YSBA332), clade III WT (B11221), clade III *cna1Δ* (YSBA336), clade IV WT (B11245), and clade IV *cna1Δ* (YSBA365) strains. WT and mutant strains were cultured overnight in liquid YPD medium at 30°C, serially diluted 10-fold, and spotted on YPD medium supplemented with stressors, including 3 mg/mL CFW, 0.2% SDS, 0.05 µg/mL PSC, or 0.1 µg/mL CAF. (C) Phenome heat map of *cna1Δ* mutants across four different *C. auris* clades. Phenotype scores are color-coded based on qualitative or semi-quantitative measurements under the indicated growth conditions. Abbreviations: 30T, 30°C; 37T, 37°C; 42T, 42°C; AMB, amphotericin B; FDX, fludioxonil; 5FC, 5-flucytosine; FLC, fluconazole; PSC, posaconazole; ITC, itraconazole; VRC, voriconazole; KTC, ketoconazole; CAF, caspofungin; MIF, micafungin; ANF, anidulafungin; HPX, hydrogen peroxide; TBH, tert-butyl hydroperoxide; DIA, diamide; MD, menadione; MMS, methyl methanesulfonate; HU, hydroxyurea; TM, tunicamycin; DTT, dithiothreitol; CR, Congo red; CFW, calcofluor white; SDS, sodium dodecyl sulfate; CDS, cadmium sulfate; KCR, YPD + KCl; NCR, YPD + NaCl; SBR, YPD + sorbitol; KCS, YP + KCl; NCS, YP + NaCl; SBS, YP + sorbitol. Red and blue gradients represent phenotype reduction and enhancement, respectively, with strong, intermediate and weak phenotypes indicated by color intensity. (D) Growth in Sabouraud Dextrose (SabDex) medium showing cell aggregation in the WT and *cna1Δ* strain across different clades.

PLOS Pathogens

Upon CFW treatment, *CNA1* expression was moderately upregulated across all clades, with no substantial differences observed in either basal or CFW-induced levels (S9 Fig). These results suggest that the clade-specific phenotypes are unlikely to result from differential *CNA1* expression, but rather may reflect variation in downstream components or modulators of the calcineurin pathway.

Cell aggregation is a clade-specific phenotypic trait in *C. auris*, particularly prominent in clade III, where it is recognized as a key feature influencing virulence, skin colonization, and biofilm formation [23]. To explore the role of calcineurin in this trait, we examined the aggregation ability of the *cna1*Δ mutants across different clades. Notably, the clade III and IV *cna1*Δ mutants exhibited enhanced aggregation compared to the wild-type strain, with the clade III *cna1*Δ mutant showing the most pronounced effect (Fig 3D). This enhanced aggregation might contribute to its increased resistance to CFW (Fig 3C). All these findings demonstrate that the calcineurin pathway plays both evolutionarily conserved and clade-specific divergent roles in *C. auris*.

## Crz1 is the transcription factor downstream of calcineurin in *C. auris*

To identify transcription factors downstream of calcineurin, we queried the *Candida* genome database and discovered two potential Crz1 orthologs in *C. auris*: B9J08_002096 and B9J08_000447. B9J08_002096 was designated as Crz1 because it shares a closer phylogenetic relationship with *C. albicans* Crz1 than B9J08_000447, whose homolog in *C. albicans* is Crz2 (S10A Fig). Both Crz1 and Crz2 contain conserved zinc finger $C_2H_2$ domains, which are essential for DNA binding (S10B, S10C, and S10D Fig). To elucidate the roles of Crz1 and Crz2 in *C. auris*, we generated *crz1*Δ, *crz2*Δ, and *crz1*Δ *crz2*Δ mutants using the B8441 strain as the parental strain (S11A, S11B, and S11C Fig).

The *crz1*Δ mutant displayed no growth defects at temperatures up to 42°C, similar to the calcineurin mutants (Fig 4A and 4C). However, at temperatures exceeding 42°C, the *crz1*Δ mutant exhibited impaired growth, suggesting that Crz1, like calcineurin, contributes to extreme thermotolerance (Fig 4B and 4C). Notably, the growth defect in the *crz1*Δ mutant was less pronounced than that observed in the calcineurin mutants (Fig 4B and 4C). These findings suggest that while Crz1 plays a supporting role in extreme thermotolerance, calcineurin appears to have a broader and more critical function.

The *crz1*Δ mutant displayed moderate sensitivity to fludioxonil and ER stress inducers, such as tunicamycin and DTT, compared to the calcineurin mutant (Fig 4D). Additionally, the *crz1*Δ mutant exhibited highly increased sensitivity to cell membrane stress induced by SDS, comparable to that observed in the *cnb1*Δ *cna1*Δ mutant (Fig 4D). Moreover, the *crz1*Δ mutant was more susceptible to azole drugs, including FLC, PSC, VRC, and KTC, than the wild-type strain, exhibiting phenotypes similar to the *cnb1*Δ *cna1*Δ mutant but with reduced severity (Figs 4D and S12A). Complementation with the wild-type *CRZ1* allele restored normal phenotypes in the *crz1*Δ mutant (Figs 4D and S12A). In contrast, the *crz2*Δ mutant exhibited minor phenotypic alterations (Fig 4D). Furthermore, the *crz1*Δ *crz2*Δ double mutant exhibited phenotypes identical to those of the *crz1*Δ mutant alone. All these data suggest that Crz1, but not Crz2, likely functions as the downstream effector of calcineurin in *C. auris*.

In several other fungi, activated calcineurin dephosphorylates Crz1, facilitating its translocation into the nucleus [5]. To investigate the cellular localization of Crz1 in *C. auris*, we constructed the *crz1*Δ::*CRZ1-mCherry* strain, in which the *crz1*Δ mutant was complemented with a C-terminal *mCherry*-tagged *CRZ1* allele (S13A Fig). The *CRZ1-mCherry* allele was functional, as it restored wild-type phenotypes in the *crz1*Δ mutant (S13C Fig). Upon FLC treatment for up to 4 h, the Crz1-mCherry protein predominantly localized to the nucleus after 3 h (Fig 4E and 4F). To confirm that calcineurin mediates Crz1 nuclear translocation, we deleted *CNA1* in the Crz1-mCherry tagged complemented strain (S13 Fig). In the absence of *CNA1*, Crz1 failed to localize to the nucleus in response to FLC (Fig 4E and 4F). These findings demonstrate that Crz1 functions as the downstream transcription factor of calcineurin and undergoes nuclear translocation in response to azoles in a calcineurin-dependent manner.

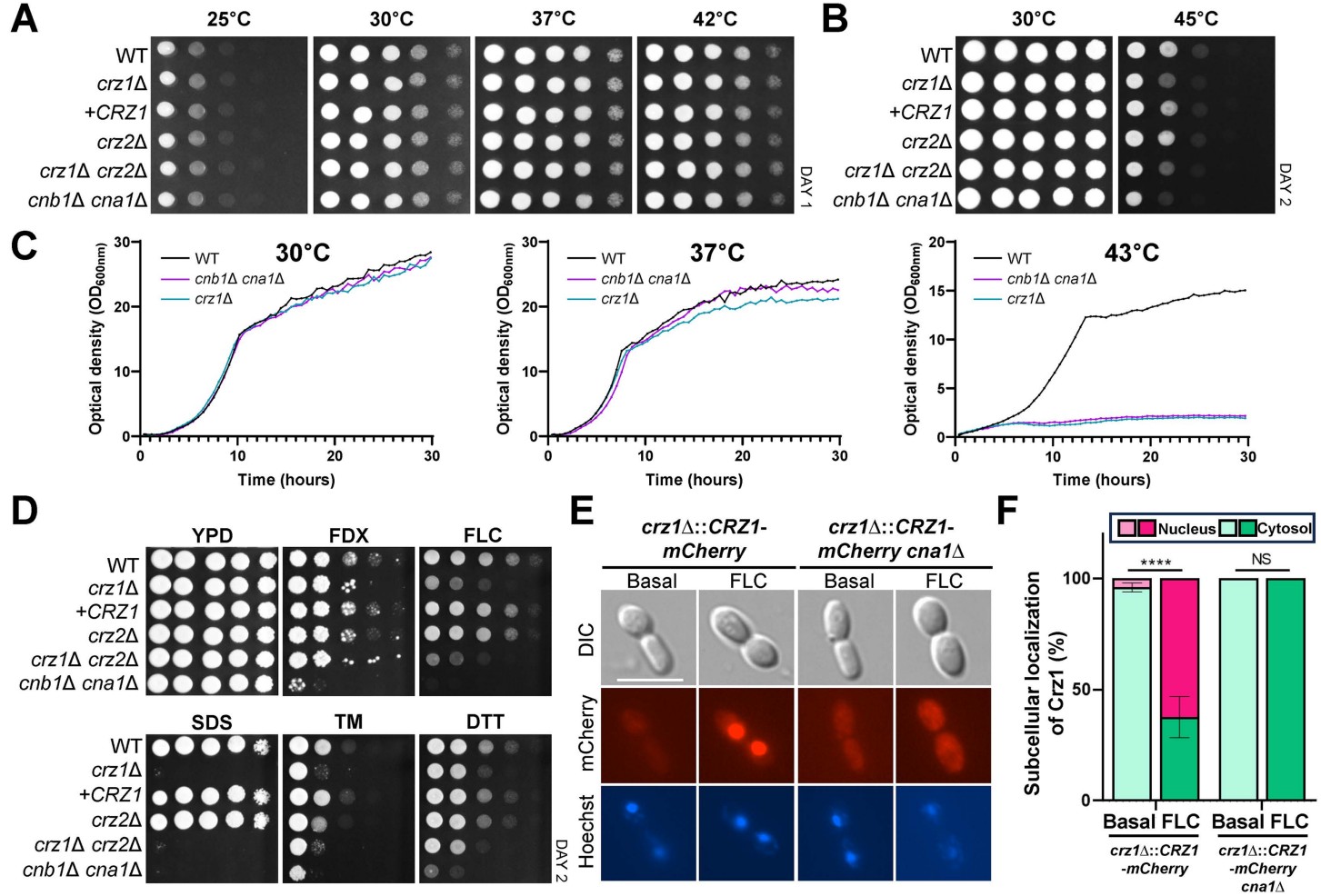

**Fig 4. Crz1 functions as the downstream transcription factor of calcineurin in *C. auris*.** (A) Qualitative spot assays display the growth of the following strains under varying temperatures: wild-type (WT; B8441), *crz1Δ* (YSBA105), *crz1Δ::CRZ1* (+*CRZ1*; YSBA158), *crz2Δ* (YSBA143), *crz1Δ crz2Δ* (YSBA153), and *cnb1Δ cna1Δ* (YSBA172). WT and mutant strains were cultured overnight in liquid YPD medium at 30°C, serially diluted 10-fold, and spotted on YPD agar plates. Plates were incubated for 1 day at 25°C, 30°C, 37°C, or 42°C. (B) WT and mutant strains were spotted on YPD plates and incubated at 30°C or 45°C for 2 days. (C) Quantitative growth rates of the WT, *cna1Δ cnb1Δ*, and *crz1Δ* strains were measured at 30°C, 37°C, and 43°C using a multichannel bioreactor (Biosan Laboratories, Inc., Warren, MI) by measuring the $OD_{600}$ for 30 h. (D) Qualitative spot assays showing the stress susceptibility of WT (B8441), *crz1Δ* (YSBA105), *crz1Δ::CRZ1* (YSBA158), *crz2Δ* (YSBA143), *crz1Δ crz2Δ* (YSBA153), and *cnb1Δ cna1Δ* (YSBA172) strains. WT and mutant strains were grown overnight in liquid YPD medium at 30°C, serially diluted 10-fold, and spotted on YPD medium containing stressors, including 3 µg/mL FDX, 150 µg/mL FLC, 0.1% SDS, 3.6 µg/mL TM, and 22 mM DTT. Plates were incubated for 2 days at 30°C. (E) Nuclear translocation of Crz1-mCherry upon FLC treatment. Cells were cultured overnight, synchronized to an $OD_{600}$ of 0.2, and further grown until an $OD_{600}$ of 0.8 was reached. Cells were treated with 400 µg/mL FLC for 3 h, harvested (1 mL), fixed, stained with Hoechst, and imaged using a fluorescence microscope. The white scale bar represents 10 µm. (F) Bar chart quantifying the proportion of cells exhibiting Crz1-mCherry localization in either the nucleus or cytoplasm. A total of 50 cells were counted (n = 50). Statistical significance was evaluated using two-way ANOVA Analysis. Data are presented as mean ± SEM (NS, no significant; ****, $P < 0.0001$).

## Crz1 and calcineurin play distinct roles in resistance to cell wall-damaging stressors and echinocandins in *C. auris*

We identified an unexpected phenotypic difference between the *crz1Δ* and calcineurin mutants. The *crz1Δ* mutant exhibited markedly increased resistance to cell wall-damaging stressor CFW and echinocandins, including CAF, MIF, and ANF, which was in stark contrast to the heightened susceptibility observed in the *cna1Δ* and *cnb1Δ* mutants (Figs 5A and S12).

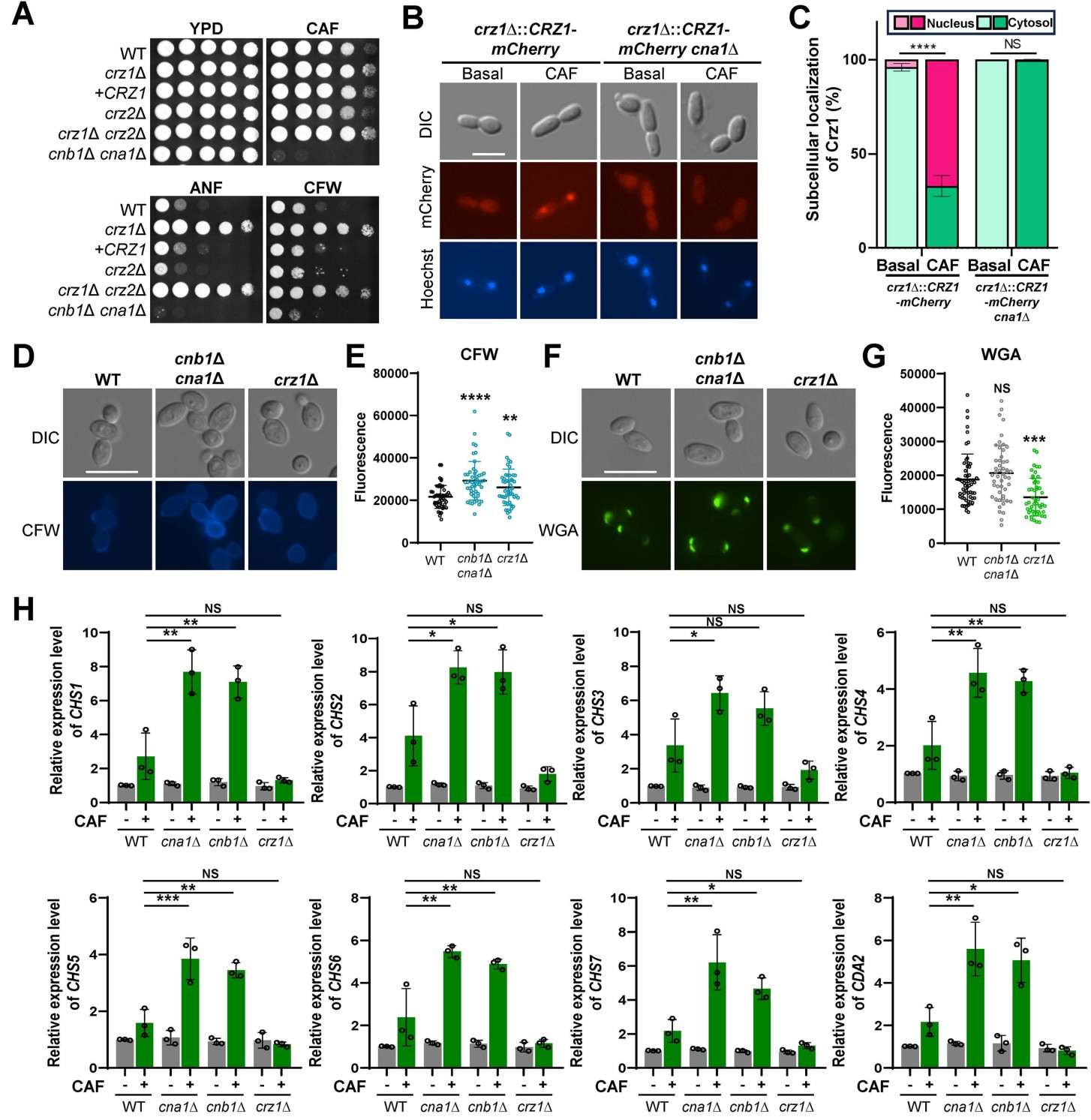

**Fig 5. Crz1 opposes calcineurin in regulating echinocandin resistance in *C. auris*.** (A) Spot assays showing the growth of WT and mutant strains on YPD medium containing 2.5 μg/mL CAF, 1 μg/mL ANF, or 3 mg/mL CFW. Plates were incubated for 2 days at 30°C. (B) Nuclear translocation of Crz1-mCherry upon treatment with 10 μg/mL CAF for 30 min. (C) Bar chart quantifying the proportion of cells exhibiting Crz1-mCherry localization in either the nucleus or cytoplasm. A total of 50 cells were analyzed (*n* = 50). Statistical significance was determined using two-way ANOVA. Data are shown as

mean ± SEM (NS, no significant; ****, $P < 0.0001$). (D, F) Chitin and chito-oligomers staining in the WT and mutant strains. Cells were grown overnight at 30°C in liquid YPD medium, subcultured to an $OD_{600}$ of 0.8, and stained with wheat germ agglutinin (WGA) or CFW. Representative fluorescence microscopy images are displayed for each strain (scale bar, 10 µm). (E, G) Quantitative fluorescence measurements of at least 50 individual cells per strain, analyzed using ImageJ/Fiji software, are shown. (H) qRT-PCR analysis of chitin synthase genes (CHS1-CHS7) and chitin deacetylase gene (CDA2) in WT, cna1Δ, cnb1Δ, and crz1Δ strains. Cells were cultured for 24 h with 5 µg/mL CAF at 30°C in a shaking incubator. Gene expression levels were normalized to ACT1, and fold changes were calculated relative to the basal expression level in WT. (E, G, H) Statistical significance was evaluated using one-way ANOVA with Bonferroni's multiple-comparison test (*, $P < 0.05$; **, $P < 0.01$; ***, $P < 0.001$; ****, $P < 0.0001$; NS, not significant).

Interestingly, Crz1 was able to undergo nuclear translocation in response to CAF in a calcineurin-dependent manner (Fig 5B and 5C). These seemingly contradictory findings suggest that Crz1 may act as both a transcriptional activator and repressor, depending on the external cues and environmental context. Alternatively, Crz1 may regulate other transcription factors that, in turn, modulate genes involved in maintaining cell wall integrity in *C. auris*.

These findings suggest that calcineurin and Crz1 may play distinct roles in maintaining fungal cell wall integrity, as CFW and echinocandins disrupt chitin and β-glucan layers, respectively, which are major components of the fungal cell wall [24]. To further investigate the differential roles of calcineurin and Crz1 in maintaining cell wall integrity, we analyzed CFW and wheat germ agglutinin (WGA) staining patterns in the wild-type, calcineurin (cnb1Δ cna1Δ), and crz1Δ mutant strains. CFW binds to β-linked polysaccharides, such as chitin and chitosan, while WGA with high molecular weight specifically binds to surface-exposed chito-oligomers [25,26]. CFW staining revealed increased fluorescence in both calcineurin and crz1Δ mutants compared to the wild type, indicating enhanced chitin accumulation in these mutants (Fig 5D and 5E). However, WGA staining showed that fluorescence levels in the calcineurin mutants were similar to those of the wild type, whereas the crz1Δ mutant exhibited significantly reduced fluorescence (Fig 5F and 5G). These results suggest that deletion of CRZ1, but not calcineurin, reduces chito-oligomer exposure.

To further investigate the roles of calcineurin and Crz1 in chitin biosynthesis, we measured the expression levels of the chitin synthase (CHS) and chitin deacetylase (CDA) genes. Under basal conditions, no significant changes in expression levels were observed (Fig 5H). However, upon treatment with CAF, the expression levels of chitin synthase genes (CHS1-CHS7) and CDA2 were significantly upregulated in the cna1Δ or cnb1Δ mutants compared to the wild-type strain, whereas the crz1Δ mutant showed no notable induction (Fig 5H). These findings suggest that calcineurin mutants compensate for this heightened echinocandin susceptibility by upregulating chitin synthesis genes to reinforce the cell wall. In contrast, the crz1Δ mutant, which exhibits resistance to echinocandins, did not induce chitin synthesis under these conditions (Fig 5H). These findings underscore the complexity of the calcineurin/Crz1 pathway in maintaining fungal cell wall integrity.

To assess whether the induction of CHS genes in the cna1Δ mutant upon CAF exposure is mediated by the PKC-dependent cell wall integrity pathway, we generated a cna1Δ mkc1Δ double mutant by deleting MKC1 (B9J08_002682), a mitogen-activated protein kinase (MAPK) central to the PKC pathway in *C. albicans* [27], in the cna1Δ background. Quantitative gene expression analysis revealed that CHS gene upregulation was further enhanced in the cna1Δ mkc1Δ mutant compared to the cna1Δ single mutant following CAF treatment (S14 Fig). These results demonstrate that the increased CHS expression in the absence of calcineurin is not positively regulated by the PKC-Mkc1 MAPK cascade. Instead, they implicate the involvement of additional PKC-independent signaling pathways that are activated in response to cell wall stress when calcineurin function is impaired.

## The calcineurin pathway is dispensable for morphogenesis, biofilm formation, secreted aspartyl protease activity, and ploidy switching in *C. auris*

We next examined whether the calcineurin/Crz1 pathway influences other known virulence traits in *C. auris*. Morphological changes, critical virulence factors in *Candida* species, were assessed by inducing pseudohyphal formation with the DNA-damaging agent hydroxyurea [28]. Both calcineurin and crz1Δ mutants exhibited no significant differences in pseudohyphal formation compared to the wild type (S15A Fig). Biofilm formation and the production of secreted aspartyl

proteases (SAPs) were also analyzed in these mutants. Our findings confirmed that the calcineurin/Crz1 pathway does not play a role in either process (S15B and S15C Fig).

In addition, we investigated the frequency of ploidy switching, another virulence trait in *C. auris* [29]. It has been reported that haploid and diploid *C. auris* cells grown on YPD medium containing phloxine B display white and pink colony phenotypes, respectively [29]. Interestingly, most calcineurin mutants, but not the *crz1Δ mutant*, formed pink colonies (S15D Fig). However, fluorescence-activated cell sorting (FACS) analysis revealed that the pink-colored calcineurin mutant cells were not diploid (S15D Fig), suggesting that ploidy switching did not occur in these strains. We hypothesized that the pink colony phenotype may result from altered cell membrane or wall composition, which allows the dye to penetrate more easily in the calcineurin mutant strains compared to the wild-type strain. Collectively, these results demonstrate that the calcineurin/Crz1 pathway is not involved in morphogenesis, biofilm formation, secreted aspartyl protease activity, and ploidy switching in *C. auris*.

## Roles of the calcineurin pathway in the pathogenicity of *C. auris*

To determine the role of the calcineurin pathway in the pathogenicity of *C. auris*, we first utilized the *Drosophila melanogaster* infection model, a well-established fungal infection system, recognized for its short lifecycle, cost-effectiveness, ease of handling, and genetic consistency [30]. A total of 80 flies per group were injected with wild-type, *cna1Δ*, *cna1Δ::CNA1*, or *bcy1Δ* strains, and the survival rates were monitored over four days (Fig 6A). The *bcy1Δ* (YSBA4) strain, which lacks the regulatory subunit of protein kinase A (PKA) in the cAMP pathway, was included as a control strain due to its previously reported attenuated virulence in murine systemic infection models [31]. Flies infected with the *cna1Δ* mutants exhibited a significantly higher survival rate than those infected with the wild-type and *cna1Δ::CNA1* strains. However, this increased survival was less pronounced than in flies infected with the *bcy1Δ* control strain (Fig 6B). These results indicate that the calcineurin pathway is required for the virulence of *C. auris* in the insect infection model.

To determine whether the attenuated virulence observed in the *cna1Δ* mutant is mediated through Crz1, a downstream transcription factor of calcineurin, we assessed the virulence of the *crz1Δ* mutant using the same *Drosophila* infection model. Notably, the survival rates of flies infected with the *crz1Δ* mutant were comparable to those infected with the wild-type strain, indicating that loss of *CRZ1* does not affect virulence in this model (Fig 6C). These findings suggest that the calcineurin-dependent regulation of *C. auris* virulence is largely Crz1-independent and may involve alternative downstream targets or pathways.

Next, an ex vivo skin adhesion model was utilized to evaluate the ability of *C. auris* to attach to mammalian skin tissue, a critical step in its colonization and infection process. Adhesion of each strain to murine ear skin tissue was quantified, and the *cna1Δ* mutant showed significantly reduced adhesion at the early 1-h time point compared to the wild-type and complemented strains. However, no significant difference was observed after 2 h of incubation (Fig 6E). These results suggest that calcineurin signaling is particularly important during the early stage of *C. auris* tissue attachment.

We next evaluated the subcutaneous and systemic infectivity of the calcineurin mutants using a murine (BALB/c) infection model. The fungal burdens in infected skin regions were quantified 9 days post-infection (dpi). Mice infected with the *cna1Δ* mutant exhibited significantly reduced abscess size and fungal burden compared to those infected with the wild-type strain (Fig 6H, 6I and 6J). In contrast, mice infected with the *crz1Δ* mutant showed no significant differences in fungal burden relative to the wild type (Fig 6J). These results highlight the critical role of Cna1 as a virulence factor necessary for *C. auris* skin colonization, whereas Crz1 appears dispensable in this context.

Notably, the calcineurin mutant displayed a distinct virulence pattern in a systemic murine infection model compared to the skin infection model. Mice intravenously infected with the *cna1Δ* mutant exhibited survival rates similar to those infected with the wild-type or *cna1Δ::CNA1* strain (Fig 6K). However, noticeable body weight loss in *cna1Δ*-infected mice became apparent after 8 dpi, in contrast to mice infected with the wild-type or *cna1Δ::CNA1* strains (Fig 6L). In a systemic infection model treated with FLC and CAF, the fungal burden of the *cna1Δ* mutant in the inner ear was significantly more

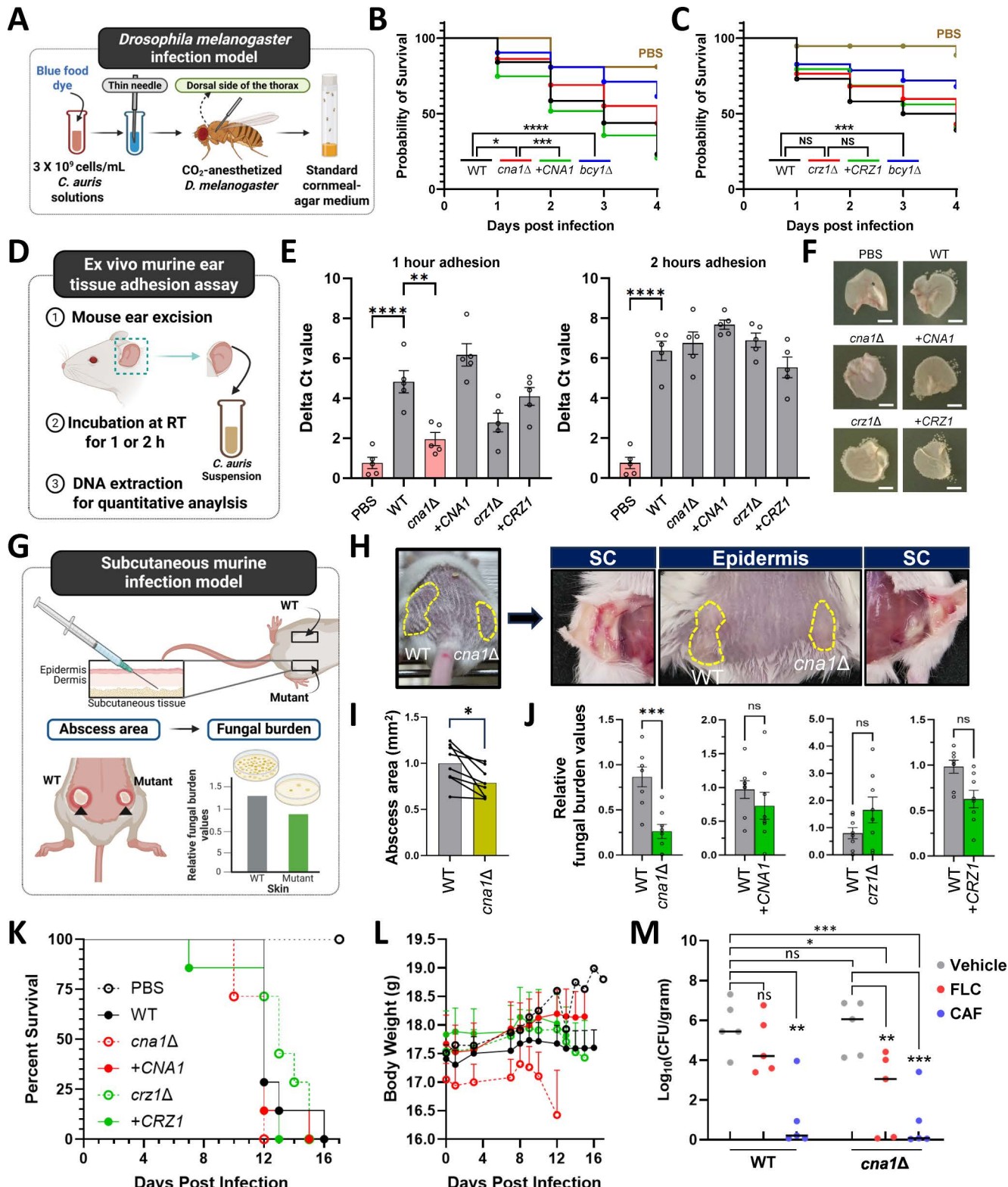

**Fig 6. The role of the calcineurin pathway in the pathogenicity of *C. auris*.** (A) Schematic representation of the experiment processes for the *Drosophila melanogaster* infection model. This figure was created using BioRender (https://biorender.com/). (B) Survival rates of *D. melanogaster* infected with WT, *cna1Δ*, +*CNA1*, or *bcy1Δ* strains were monitored over 4 days post-infection (dpi). A total of 80 files were used per strain. Survival curves were

statistically analyzed using the log-rank (Mantel-Cox) test. The PBS control is represented in brown. (C) Survival rates of *D. melanogaster* infected with WT, *crz1Δ*,+*CRZ1*, or *bcy1Δ* strains. (D) Schematic diagram depicting an *ex vivo* mouse ear infection model. This figure was generated using BioRender (https://biorender.com/). (E) Equal-sized ex vivo mouse ear tissues (n = 6) were incubated with PBS-diluted *C. auris* cells on a rotator to allow adhesion. After incubation, adherent cells were gently scraped off, and genomic DNA was extracted. Quantitative PCR targeting the *CNB1* gene was performed, and Ct values were normalized to those of the non-infected PBS-treated group. (F) After 1-h incubation, ear tissues were placed on YPD agar plates and incubated at room temperature for 2 days, followed by imaging. Scale bar indicates 0.5 mm. (G) Schematic diagram of the mouse skin infection model. This figure was designed using BioRender (https://biorender.com/). (H) A total of $10^7$ cells from WT or mutant strains were subcutaneously injected into the left and right flanks of 7-week-old BALB/c mice (n = 8). (I) Clinical scores were determined by measuring abscess areas. Statistical significance was calculated using an unpaired *t*-test (*, $P = 0.0424$). (J) Fungal burden was assessed by homogenizing abscess tissue 9 dpi and plating the homogenate on YPD agar plate. Relative fungal burden values were calculated as mutant CFU/g per WT CFU/g. Statistical significance was calculated using an unpaired *t*-test (***, $P < 0.0005$). (K, L) Survival curve and body weight changes of 7-week-old BALB/c mice (n = 7) intravenously injected with $10^7$ *C. auris* cells. (M) Fungal burden in the inner ear was measured 7 days post-intravenous systemic infection. Mice were treated with FLC (20 mg/kg) and CAF (2 mg/kg) via intraperitoneal injection, administered once daily for six days, starting on the day of infection. Statistical significance was calculated using an unpaired *t*-test (**, $P < 0.01$; ***, $P < 0.001$).

reduced by FLC compared to the wild-type strain (Fig 6M). Collectively, these findings suggest that the calcineurin pathway plays distinct roles in the virulence of *C. auris*, depending on the specific host niche.

## Discussion

This study investigates the critical role of the calcineurin signaling pathway in *C. auris*, emphasizing its essential functions in maintaining cell membrane and cell wall integrity, mediating antifungal resistance, facilitating environmental stress adaptation, and contributing to pathogenicity (summarized in Fig 7). Calcineurin deletion (*cna1Δ* and *cnb1Δ*) results in pronounced hypersensitivity to high temperatures (>43°C) and cell membrane- and cell wall-disrupting agents. Loss of calcineurin markedly increases susceptibility to azoles and echinocandins, while enhancing resistance to AMB, underscoring its potential as a therapeutic target. This is further reinforced by the synergistic antifungal effects observed when calcineurin inhibitors, such as FK506 or CsA, are combined with echinocandins or azoles. Crz1, the downstream transcription factor of calcineurin in *C. auris*, plays a central role in mediating responses to membrane and ER stress and in modulating drug susceptibility. Deletion of *CRZ1* increases susceptibility to azoles and membrane stressors but enhances resistance to echinocandins, suggesting that Crz1's functions oppose certain aspects of calcineurin's role. While calcineurin is dispensable for systemic infection in a mammalian host, it plays an important role in skin colonization, further supporting its potential as a universal antifungal target against multidrug-resistant fungal pathogens.

In *C. auris*, the calcineurin pathway shares several conserved roles with other *Candida* species (*C. albicans*, *C. glabrata*, *Candida tropicalis*, and *Candida dubliniensis*) as well as major fungal pathogens such as *Aspergillus fumigatus* and *C. neoformans*. These roles include maintaining cell membrane and wall integrity, mediating resistance to environmental stresses, and promoting antifungal drug resistance, particularly to azoles and echinocandins. However, *C. auris* exhibits unique species-specific characteristics. For instance, in *C. neoformans*, calcineurin controls sexual differentiation [12]. In *A. fumigatus*, calcineurin controls normal filamentous growth [32]. Furthermore, in *C. dubliniensis* and *C. tropicalis*, calcineurin plays a crucial role in hyphal growth [9]. In contrast, our findings reveal that the calcineurin pathway in *C. auris* does not regulate morphogenic transitions. Moreover, calcineurin promotes thermotolerance at 37°C in *C. glabrata* and *C. neoformans* [6,10]. However, in *C. auris*, calcineurin is not required for thermotolerance at 37°C but is specifically essential for survival at extreme temperatures exceeding 43°C, underscoring its critical role in high-temperature adaptation. These results highlight both the conserved and distinct roles of the calcineurin pathway in *C. auris*, underscoring the pathogen's specialized adaptation to its unique niche and stress environments.

The thermotolerance of *C. auris*, particularly its ability to survive and grow at temperatures ≥42°C, may play a pivotal role in its environmental persistence and transmission. As global temperatures rise due to climate change, new ecological niches with elevated temperatures are emerging. The ability of *C. auris* to survive in such warmer environments may enhance its ability to colonize new habitats, increasing the likelihood of interaction with human hosts and facilitating its

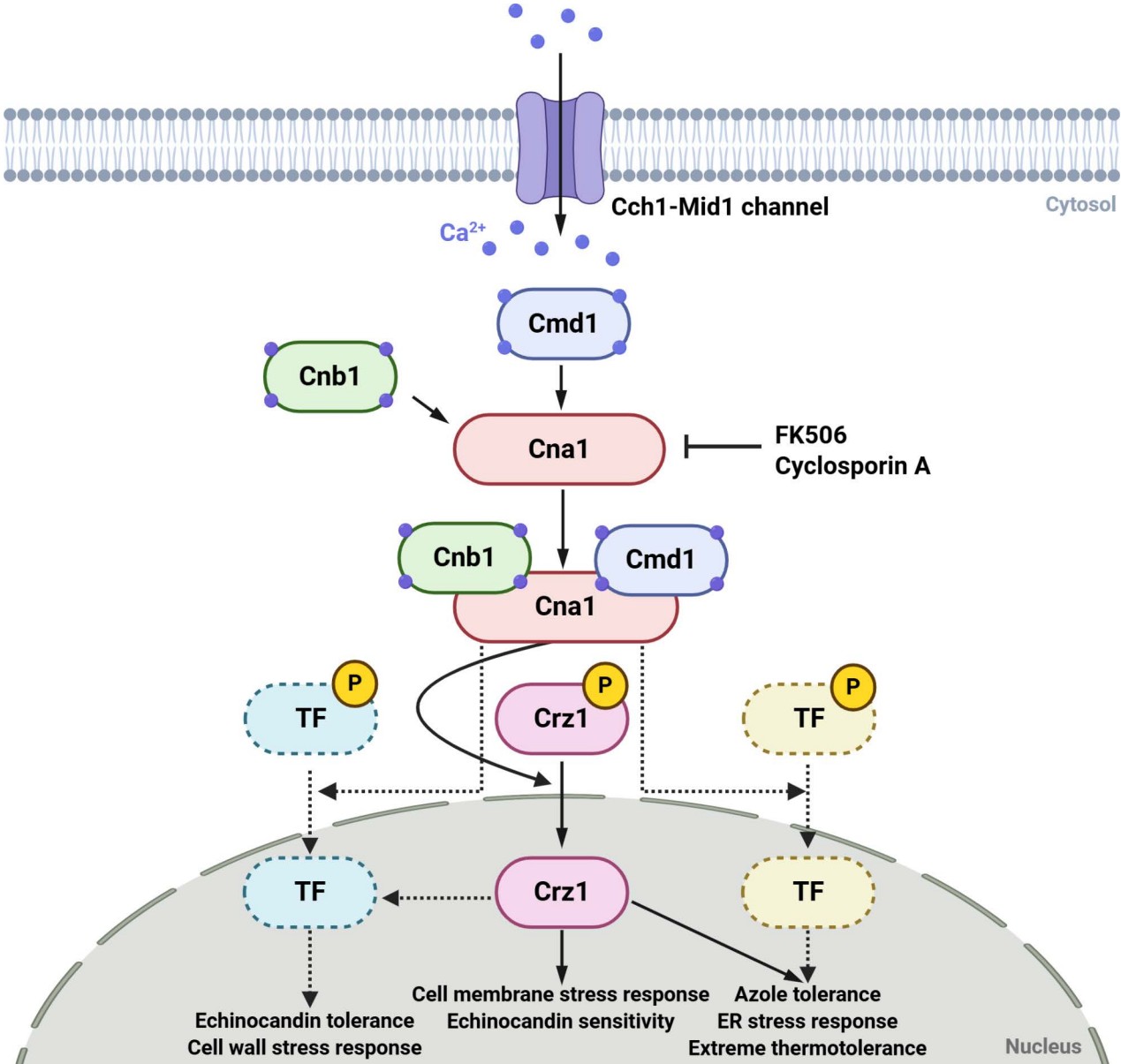

**Fig 7. Proposed regulatory mechanisms of the calcineurin pathway in *C. auris*.** The calcineurin pathway, consisting of the catalytic subunit Cna1 and the regulatory subunit Cnb1, regulates extreme thermotolerance, cell membrane and wall integrity, antifungal resistance, and virulence in *C. auris*. Upon calcium influx, activated calcineurin dephosphorylates Crz1, which translocates to the nucleus to modulate gene expression. While the calcineurin pathway promotes azole resistance, it exhibits antagonistic interactions with Crz1 in the regulation of echinocandin resistance and cell wall integrity maintenance. These findings underscore both the conserved and unique regulatory roles of the calcineurin pathway in *C. auris*. This figure was created through BioRender (https://biorender.com/).

spread. This exceptional ability to adapt to and persist in high-temperature conditions may have been a key factor in its emergence as a globally distributed pathogen.

Another unique feature of the *C. auris* calcineurin pathway is the regulatory mechanism of its downstream transcription factor Crz1. In *C. auris*, Crz1 functions downstream of calcineurin to mediate stress and drug responses through nuclear

translocation. Crz1 plays a critical role in maintaining membrane integrity and regulating azole susceptibility, as its deletion phenocopies the effects observed in calcineurin mutants. Similarly, in *C. neoformans*, calcineurin regulates stress survival and virulence through Crz1, along with additional downstream targets, including Lhp1, Pbp1, and Puf4, which contribute to post-transcriptional regulation and stress adaptation [12]. However, a distinct feature in *C. auris* is the opposing effects of calcineurin and Crz1 on echinocandin resistance: calcineurin deletion increases echinocandin susceptibility, whereas Crz1 deletion enhances resistance. This phenotype is highly unusual and contrasts with other fungal species, such as *C. albicans* and *C. glabrata*, where both calcineurin and Crz1 are required for echinocandin resistance [10,33]. These findings suggest that Crz1 may function as both an activator and a repressor, regulating distinct gene sets that influence echinocandin resistance. Alternatively, calcineurin may activate additional transcription factor(s) to promote echinocandin resistance. Although we identified Crz2, a structural paralog of Crz1, its deletion did not affect any calcineurin-dependent phenotypes. Further investigation is warranted to identify additional calcineurin-dependent transcription factors beyond Crz1 in *C. auris*.

Our findings further demonstrate that calcineurin and Crz1 play distinct roles in regulating cell wall composition in *C. auris*. Both the *cnb1Δ cna1Δ* and *crz1Δ* mutants exhibited increased CFW staining, indicating elevated accumulation of chitin and its derivative. However, the *crz1Δ* mutants showed reduced WGA fluorescence staining compared to the wild-type and calcineurin mutants. Given that WGA is a high molecular weight lectin that specifically stains surface-exposed chitins, such as chito-oligomers, deletion of *CRZ1* deletion may result in diminished surface exposure of chito-oligomers. This reduction in surface-exposed chito-oligomers could be attributed to the increased beta-glucan thickness observed in the *crz1Δ* mutants, which may also explain the enhanced echinocandin resistance in these mutants. These observations indicate that Crz1-mediated regulation of cell wall composition likely influences both chitin exposure and echinocandin resistance. Further investigations are required to elucidate differential roles of calcineurin and Crz1 in modulating cell wall architecture and composition in *C. auris*.

The conservation of calcineurin functions across *C. auris* clades highlights its critical role in stress adaptation and survival in this fungal pathogen. Phylogenetic analysis confirmed that Cna1 is highly conserved, while functional assays demonstrated consistent stress response and antifungal drug resistance phenotypes across clades. These findings underscore calcineurin as an evolutionarily conserved regulator and a promising antifungal target effective across the genetically diverse *C. auris* clades. Interestingly, a clade-specific variation in calcineurin function was observed in sensitivity to CFW. While clades I and II displayed enhanced sensitivity, clades III and IV showed reduced sensitivity. Since CFW is a cell wall-destabilizing agent that binds to chitin and its derivative, these results suggest that the calcineurin pathway may have differential roles in maintaining cell wall integrity across clades. This variation may correlate with cell aggregation phenotype predominantly observed in clade III strains [25], implying potential differences in cell surface structures among *C. auris* clades. Further investigation into calcineurin-dependent clade-specific traits is warranted to understand the underlying mechanisms driving these phenotypic differences.

We observed a notable difference in the virulence potential of calcineurin across varying animal model systems. The *cna1Δ* mutant exhibited reduced virulence in the *Drosophila* infection model. Furthermore, the *cna1Δ* mutant displayed reduced infectivity in both the ex vivo murine ear tissue adhesion assay and the subcutaneous skin infection model. In contrast, the *cna1Δ* mutant retained virulence comparable to the wild-type strain in a systemic murine infection model. Interestingly, mice intravenously infected with the *cna1Δ* mutant experienced significant body weight loss during infection, a phenomenon not observed in mice infected with the wild-type or its complemented strains. As Crz1 was dispensable for virulence in all tested animal models, we hypothesized that the opposing roles of calcineurin and Crz1 in maintaining cell wall integrity could influence virulence outcomes. This finding contrasts with the established role of calcineurin in systemic infections caused by other fungal pathogens, such as *C. albicans* and *C. neoformans*, where it is essential for adaptation to host environments and virulence [34,35]. We speculate that the altered cell wall integrity in the *cna1Δ* mutant may impair early host tissue adhesion and colonization by *C. auris*, while triggering an abnormal immune response during

systemic infection, contributing to a significant body weight loss. A similar immune response phenotype has been reported in *rim101* mutants of *C. neoformans* [36]. Further studies are required to elucidate how altered calcineurin signaling in *C. auris* impacts host immune systems during systemic infection.

Based on our data, calcineurin inhibitors may not be useful for treating systemic infection caused by *C. auris* but could prove effective for managing its skin infections or other superficial or cutaneous infections. Our data demonstrated that calcineurin inhibitors, such as FK506 and CsA, synergize with azoles and echinocandins to confer fungicidal activity against *C. auris*. Notably, we observed a significant reduction in the fungal burden of the *cna1Δ* mutant in the inner ear by FLC treatment, suggesting that the combination of azoles and calcineurin inhibitors could exhibit synergistic effects *in vivo* for treating *C. auris* infections in the inner ear. In *C. albicans*, FK506 and CsA improve FLC efficacy by impairing stress adaptation mechanisms [37,38]. Similarly, in *C. neoformans* and *A. fumigatus*, FK506 enhances azole and echinocandin activity by disrupting stress tolerance and cell wall remodeling [39,40]. In *C. glabrata*, FK506 synergizes with FLC by impairing stress adaptation [41]. Despite its potent antifungal activity, FK506 is limited by its immunosuppressive effects, prompting the development of analogs designed to reduce immunosuppression while selectively targeting fungal cells [42,43]. These FK506 analogs have demonstrated synergistic effects with FLC in *C. albicans* and *C. neoformans* [43]. While FK506's immunosuppressive properties remain a challenge, our findings underscore the potential of calcineurin inhibitors as adjunctive agents to enhance the efficacy of azoles and echinocandins for *C. auris* infections, particularly in non-systemic settings.

Interestingly, the *cna1Δ* and *cnb1Δ* mutants exhibited increased resistance to AMB while showing heightened susceptibility to azoles. This seemingly paradoxical pattern has been observed in other fungal species and reflects the fundamentally distinct mechanisms of action of these two antifungal drug classes. Azoles inhibit Erg11 enzymatic activity, thereby blocking ergosterol biosynthesis, while AMB binds directly to ergosterol in the membrane, forming lethal pores. As previously reported in *C. neoformans*, disruption of transcription factors regulating *ERG11* expression often leads to opposite susceptibility to FLC and AMB. For instance, *sre1Δ* mutants, where *SRE1* encodes the central sterol-regulating TF [44], exhibit increased resistance to AMB but greater sensitivity to FLC [45]. In line with this, *ERG11* expression was moderately increased in the calcineurin mutants following FLC treatment, which may contribute to FLC susceptibility. However, ergosterol quantification showed no significant differences between wild-type and calcineurin mutant strains, suggesting that ergosterol abundance does not account for the observed AMB resistance. To elucidate the underlying mechanisms by which calcineurin deletion confers AMB resistance, transcriptome analyses of wild-type and *cna1Δ* strains under both untreated and AMB-treated conditions will be necessary. These analyses will enable the identification of calcineurin-regulated gene networks and downstream effectors implicated in AMB resistance, potentially uncovering novel pathways involved in cell wall and membrane remodeling, stress adaptation, and antifungal tolerance that compensate for the loss of calcineurin signaling.

In conclusion, our study demonstrates that the calcineurin complex and its downstream transcription factor Crz1 play overlapping and distinct roles in regulating extreme thermotolerance, cell membrane and wall integrity, antifungal drug resistance, and virulence in *C. auris*. Further genetic, transcriptomic, and biochemical investigations are warranted to elucidate the precise mechanisms underlying the calcineurin complex and its downstream regulatory network in *C. auris*.

## Materials and methods

### Ethics statement

All animal care and experimental procedures were reviewed and approved by the Institutional Animal Care and Use Committee of the Experimental Animal Center at Jeonbuk National University (Approval number JBNU 2023–131). All experiments were conducted in full compliance with established ethical guidelines for animal research.

### *Candida auris* strains and growth media

The *Candida auris* strains used in this research are listed in S1 Table in the supporting information. The parental wild-type strains, including B8441 (clade I, AR0387), B11220 (clade II, AR0381), B11221 (clade III, AR0383), and B11245 (clade

IV, AR0386), were provided by the Centers for Disease Control and Prevention (CDC). Both wild-type and mutant strains were maintained as frozen stocks in 20% glycerol at −80°C until use. Yeast strains were cultured on YPD agar plates (2% agar in YPD broth: 2% peptone, 1% yeast extract, and 2% D-glucose) at 30°C for 24–48 h. For liquid culture preparation, yeast cells were grown in YPD media at 30°C with constant agitation at 200 rpm. Before experimental assays, cells were transferred to fresh YPD broth and cultured to the mid-log phase ($OD_{600}$ = 0.6–0.8) prior to the designated treatments.

## Gene deletion and complementation

Gene deletion mutants were constructed using the nourseothricin resistance (*CaNAT*), hygromycin B resistance (*CaHYG*), or G418 resistance (*CaNEO*) markers, each flanked by 0.5–0.7 kb 5′ and 3′ regions of the target genes, including *CNA1, CNB1, CRZ1,* and *CRZ2*. Gene disruption cassettes containing the selection markers were assembled through double-joint PCR. In the first round of PCR, the flanking regions of each target gene were amplified using the L1-L2 and R1-R2 primer pairs. The *CaNAT* selection marker was amplified from the plasmid pV1025, which contains the *CaNAT* gene, using the appropriate primer pairs listed in S1 Table in the supporting information. Similarly, the *CaNEO* selection marker was amplified from the plasmid pTO149 RFP-NEO, which carries the *CaNEO* gene, using corresponding primer pairs. The PCR products of the flanking regions and selection markers were purified and subsequently used as templates for a second round of double-joint PCR. During this step, the 5′- and 3′-gene disruption cassettes containing split selection markers were amplified using the L1-split primer 2 and R2-split primer 1, respectively (see S1 Table in the supporting information).

Gene disruption cassettes were introduced into *C. auris* using a modified lithium acetate/heat-shock transformation protocol. Overnight cultures were grown at 30°C in 50 mL of YPD broth with constant shaking. A 1.2 mL aliquot of cultured cells was harvested by centrifugation, washed sequentially with $dH_2O$ and lithium acetate buffer (100 mM lithium acetate, 10 mM Tris, 1 mM EDTA, pH 7.5), and resuspended in 300 μL of lithium acetate buffer. For each transformation, the reaction mixture consisted of 10 μL of denatured salmon sperm DNA (Sigma, cat no. D9156), 100 μL of the cells, 500 μL of 50% PEG4000 (Sigma, cat no. P4338), and 50 μL of the amplified gene deletion cassette. The mixture was incubated at 30°C for 6 h with intermittent inverting, followed by a 20-min heat shock at 42°C and immediate cooling on ice for 1 min. The cells were then pelleted, resuspended in 1 mL of YPD medium, and incubated at 30°C for 1–2 h with shaking to allow recovery. After recovery, the cells were washed twice with fresh liquid YPD medium and plated onto selective YPD agar containing 400 μg/mL nourseothricin, 1.8 mg/mL hygromycin B, or 2.4 mg/mL G418. Plates were incubated at 37°C for 2–3 days to select transformants. Positive transformants resistant to nourseothricin, hygromycin B, or G418 were verified for the correct genotype using diagnostic PCR and confirmed via Southern blot analysis.

To validate the phenotypes of the *cna1*Δ, *cnb1*Δ, and *crz1*Δ mutants, we generated corresponding complemented strains by re-integrating the respective wild-type genes into their native loci (*cna1*Δ::*CNA1, cnb1*Δ::*CNB1*, and *crz1*Δ::*CRZ1*). Full-length gene fragments were amplified by Pfu-PCR using genomic DNA extracted from the wild-type B8441 strain as a template, along with the primer pairs listed in S2 Table in the supporting information. The amplified fragments were cloned into the TOPO vector (Invitrogen), resulting in the plasmids pTOP-CNA1, pTOP-CNB1, and pTOP-CRZ1. After sequence verification, each plasmid was modified by subcloning the *CaHYG* selection marker into the corresponding pTOP vector, generating pTOP-CNA1-HYG, pTOP-CNB1-HYG, and pTOP-CRZ1-HYG. For re-integration into the native locus, pTOP-CNA1-HYG, pTOP-CNB1-HYG, and pTOP-CRZ1-HYG were linearized using PmlI, BsmI, and BglII, respectively, and introduced into the corresponding mutant strains using the lithium acetate heat-shock transformation protocol. The successful generation of complemented strains was confirmed through diagnostic PCR, verifying the correct integration of the wild-type alleles.

## Total RNA preparation and quantitative RT-PCR

Total RNA was extracted from *C. auris* wild-type and calcineurin/Crz1 mutant strains cultured overnight at 30°C in YPD broth. Cells were harvested by centrifugation at mid-log phase ($OD_{600}$ of 0.6–0.8), rapidly frozen in liquid nitrogen, and

lyophilized. For RNA isolation, 30 mL of the culture was maintained as the basal condition, while the remaining 30 mL was subjected to additional incubation with specific stress agents. Total RNA was extracted using the Trizol-based Easy Blue reagent (Intron). Complementary DNA (cDNA) was synthesized from purified total RNA using reverse transcriptase (Thermo Scientific) following the manufacturer's protocol. Quantitative PCR (qPCR) was performed using a CFX96 Real-Time system (Bio-Rad) with gene-specific primer pairs. Expression levels were normalized to the housekeeping gene *ACT1*. Statistical analyses were conducted using one-way ANOVA followed by Bonferroni's multiple-comparison test. To ensure reproducibility, all experiments were performed in technical triplicates and independently repeated three times with biological replicates.

## Growth and stress sensitivity spot assay

To evaluate the growth and stress sensitivity of wild-type and calcineurin/Crz1 mutant strains of *C. auris*, cells were cultured overnight at 30°C, serially diluted 10-fold up to four times (final dilution 1:10$^4$), and spotted onto YPD plates. The plates were incubated at various temperatures (25°C, 30°C, 37°C, 42°C, and 45°C), and growth was qualitatively assessed by photographing the plates after 24 h of incubation. To assess stress responses, specific chemical stressors were incorporated into the growth media, including an osmotic stressor (sorbitol), cation and salt stressors (NaCl or KCl), oxidative stressors [hydrogen peroxide (HPX), *tert*-butyl hydroperoxide (TBH), menadione (MD), or diamide (DIA)], a membrane destabilizing stressor (SDS), cell-wall destabilizing stressors [Congo red (CR) and calcofluor white (CFW)], and various antifungal agents [fludioxonil (FDX), flucytosine (5FC), fluconazole (FLC), itraconazole (ITC), posaconazole (PSC), ketoconazole (KTC), itraconazole (ITC), caspofungin (CAF), micafungin (MIF), anidulafungin (ANF) or amphotericin B (AMB)]. The plates were grown at 30°C and photographed after 2–3 days of incubation under stress conditions.

## EUCAST MIC test and checkerboard assay

Wild-type and mutant *C. auris* strains were cultured overnight at 30°C in YPD medium, washed twice with dH$_2$O, and resuspended in dH$_2$O. For the EUCAST (European Committee on Antimicrobial Susceptibility Testing) MIC assay, the cell suspension was adjusted to an OD$_{600}$ of 1.0. A total of 100 µL of the prepared cell suspension was added to 10 mL of 3-(*N*-morpholino)propanesulfonic acid (MOPS)-buffered RPMI 1640 media (pH 7.4, supplemented with 0.165 M MOPS and 2% glucose), and the mixture was dispensed into 96-well plates containing 2-fold serial dilution of the test drugs. For the checkerboard assay, the cell suspension in RPMI medium was dispensed into 96-well plates containing 2-fold serial concentrations of each drug combination. The plates were incubated at 35°C for 48 h. Cell density in each well was measured at OD$_{595}$ to determine the MIC values. Following growth assessment, cultures from each well were spotted onto YPD agar plates and incubated at 30°C for 24 h to evaluate their fungicidal effects.

## Quantification of ergosterol content using HETENE method

Ergosterol quantification was performed using a modified HETENE method. Strains were pre-cultured overnight in 50 mL of YPD medium at 30°C with shaking at 220 rpm. The overnight cultures were diluted to an OD$_{600}$ of 0.2 in fresh 50 mL YPD and sub-cultured for 3–4 h under the early exponential phage until reaching an OD$_{600}$ of 0.8–1.0. Cells were then treated with AMB (3 µg/mL) for 3 h, harvested and washed with sterile distilled water. Each pellet was resuspended in 3 mL of 25% alcoholic potassium hydroxide (prepared by dissolving 25 g of KOH in 35 mL of sterile distilled water and adjusting the volume to 100 mL with absolute ethanol) and vortexed for 1 min. The suspensions were transferred to sterile borosilicate glass screw-cap tubes (16 × 100 mm) and incubated in an 85°C water bath for 1 h, followed by cooling to room temperature. Sterols were extracted by adding 1 mL of sterile distilled water and 3 mL of *n*-heptane, followed by vigorous vortexing for 3 min. For quantification, 200 µL of the n-heptane layer was mixed with 800 µL of 100% ethanol (5-fold dilution). Absorbance was measured at both OD$_{282}$ and OD$_{230}$ using a UV spectrophotometer. Ergosterol content was calculated by the following equations: % ergosterol = [(OD$_{282}$/290) × F]/ dry pellet weight – [(OD$_{230}$/518) × F]/ dry pellet weight. F

is the ethanol dilution factor [5]. To determine the sterol composition profile, the *n*-heptane (upper) layer was transferred to a clean tube and stored at –20°C for up to 24 h. For spectrophotometric analysis, 2 mL of the sterol extract was brought to room temperature to dissolve residual sterols, and absorbance was scanned from 200 to 400 nm at 1 nm intervals using a spectrophotometer.

## Membrane permeabilization assay using flow cytometry

Membrane integrity was assessed by flow cytometry using propidium iodide (PI) staining. Strains were pre-cultured overnight in 50 mL of YPD medium at 30°C with shaking at 220 rpm. A subculture was initiated at an $OD_{600}$ of 0.2 in 60 mL of fresh YPD and incubated at 30°C with shaking at 220 rpm for approximately 3 h until the $OD_{600}$ reached 0.8. One milliliter of culture was collected from each sample and centrifuged at 3000 rpm for 3 min. Cells were washed twice with PBS and fixed in 70% ethanol (300 µL of cell suspension mixed with 700 µL of 100% ethanol) at 4°C overnight with gentle rotation to prevent clumping. Following fixation, cells were washed twice with PBS and treated with 100 µL of RNase A buffer (0.2 M Tris-HCl, pH 7.5; 20 mM EDTA) containing 1 µL of RNase A (from 10 mg/mL stock), then incubated at 37°C for 4 h. Cells were washed twice with PBS, cells were adjusted to a concentration of $1 \times 10^6$ cells/mL in PBS. PI was added to a final concentration of 50 µg/mL, and the suspension was incubated in the dark at 25°C for exactly 15 min for all strains to ensure consistent staining conditions. Finally, the cells were washed once with PBS and resuspended in 500 µL PBS for flow cytometric analysis (CytoFLEX, Beckman Coulter).

## Evaluating cell aggregation

Wild-type and mutant strains were cultured in Sabouraud Dextrose (SabDex) medium for two days, washed twice with phosphate-buffered saline (PBS), and counted. A total of $2.5 \times 10^8$ cells were suspended in 5 mL PBS in sterile medical tubes, vortexed thoroughly, and photographed 5 min after vortexing [46]. Images were captured using fluorescence microscopy.

## Observing the intracellular localization of mCherry-tagged proteins

Crz1-mCherry-tagged strains were grown overnight in YPD broth at 30°C. The overnight cultures were subcultured in YPD medium until an $OD_{600}$ of 0.8 was reached. The cells were treated with antifungal drugs and incubated under the designated conditions. For cell fixation, the samples were treated with a 4% paraformaldehyde solution containing 3.4% sucrose for 15 min at room temperature. The fixed cells were thoroughly washed with a buffer containing 0.1 M $KPO_4$ and 1.2 M sorbitol. The nucleus was visualized by staining the cells with 10 µg/mL Hoechst 33342 (Thermo Fisher, USA) for 30 min in the dark. Finally, the stained cells were examined using differential interference contrast (DIC) and fluorescence microscopy (Nikon Eclipse, Japan) to determine the subcellular localization of Crz1-mCherry fusion proteins.

## Measurement of chitin and chito-oligomers content in cell walls

To measure the chitin and chito-oligomers content in the cell walls of wild-type and calcineurin/Crz1 mutant strains of *C. auris*, cells were cultured overnight at 30°C in a shaking incubator. The cells were harvested by centrifugation, washed, and resuspended in PBS (pH 7.5). For staining, the cells were treated with 100 µg/mL fluorescein isothiocyanate (FITC)-conjugated WGA or 25 µg/mL CFW for 30 min in the dark. Following staining, cells were washed three times with PBS and visualized by fluorescence microscopy (Olympus BX51). Fluorescence intensity from at least 50 individual cells per sample was quantified using ImageJ/Fiji software to determine relative chitin and chito-oligomers levels.

## Assessment of morphological transition from yeasts to pseudohyphae

To induce pseudohyphal formation in wild-type and calcineurin/Crz1 mutant strains, cells were cultured for 24 h in YPD broth supplemented with 100 mM HU. Following pseudohyphal induction, the cells were fixed using 10% formalin and

stained with 10 µg/mL Hoechst 33342 (Thermo Fisher) to visualize cellular morphology. The fixed and stained samples were incubated in the dark for 30 min, after which images were captured using fluorescence microscopy.

## Crystal violet assay for biofilm formation

*C. auris* wild-type and mutant strains were cultured overnight at 30°C in 2 mL of YPD liquid medium. The cells were washed twice with sterile $H_2O$ and resuspended in MOPS-buffered RPMI-1640 media (pH 7.4, 0.165 M MOPS and 2% glucose). The cell suspension was adjusted to an $OD_{600}$ of 0.5, and 200 µL of the suspension was dispensed into each well of a 96-well plate. The cultures were incubated at 37°C with shaking at 220 rpm for 24 h. After incubation, the cell suspensions were carefully removed, and the plates were dried in a dry oven at 65°C. Each well was then treated with 150 µL of 0.5% crystal violet solution and incubated at room temperature for 10 min for staining. Excess dye was removed by washing the wells three times with PBS, and the plates were dried again in the dry oven at 65°C. To solubilize the bound crystal violet, 200 µL of 33% acetic acid was added to each well and incubated for 2 min. The solubilized solution was diluted 1:10 with PBS and transferred to a new 96-well plate. Absorbance was measured at $OD_{595}$ using a microplate reader.

## Secreted aspartyl proteinase activity assay

SAP activity was assessed using the yeast carbon base-bovine serum albumin (YCB-BSA) method. *C. auris* strains were grown overnight in 2 mL of YPD broth at 30°C, harvested by centrifugation, washed with $dH_2O$, and resuspended in 1 mL of $dH_2O$. The cell suspension was adjusted to a final concentration of $10^5$ cells/mL. Subsequently, 3 µL of the cell suspension was spotted onto YCB-BSA plates (containing 23.4 g/L yeast carbon base and 0.2% BSA) and incubated at 37°C for 3 days. SAP activity was quantified by measuring the diameter of the halo formed around the colonies. All experiments were performed in triplicate with independent biological repeats to ensure reproducibility.

## Ploidy switching and flow cytometry analyses

*C. auris* wild-type and calcineurin/Crz1 mutant strains were cultured overnight at 30°C in YPD medium, washed twice with PBS, and counted. For the ploidy switching assay, 100 cells were spread onto YPD media supplemented with 5 µg/mL phloxine B and incubated at 25°C for 14 days. For DNA content analysis, single colonies from YPD medium containing phloxine B were inoculated into liquid YPD medium and cultured overnight at 30°C. The cells were then harvested, washed with PBS, and counted. A total of $10^6$ cells were fixed in 70% ethanol for 14–16 h at 4°C. Following fixation, cells were washed twice with PBS and treated with RNase A (200 µg/mL) at 37°C for 1 h to remove RNA contamination. After centrifugation, the supernatant was discarded, and the cells were stained with 500 µL of 100-µg/mL propidium iodide for 30 min at room temperature in the dark. The stained cells were washed with PBS, resuspended in 300 µL of PBS, and analyzed for DNA content using flow cytometry. A minimum of 10,000 cells were evaluated per experiment to ensure statistical reliability.

## *Drosophila melanogaster* infection model

The *Drosophila melanogaster* (w1118) was used for infection experiments. Flies were maintained at 25°C with a 12-h light-dark cycle on standard cornmeal-agar medium obtained from Bloomington Stock Center. Cells from *C. auris* wild-type and mutant strains were cultured overnight at 30°C in YPD medium, followed by three washes with PBS. The cell concentration was adjusted to $3 \times 10^9$ cells/mL. The cell suspensions were centrifuged, and the resulting pellets were resuspended in PBS containing 1% (w/v) blue food dye (FD&C Blue #1) to ensure visualization. For the survival assay, adult female flies (4–5 days old) were anesthetized using $CO_2$ and the dorsal thorax was pricked with a thin needle dipped in the prepared cell suspension. Approximately 80 flies per strain were infected, divided into groups of 10, and placed into

individual vials containing standard cornmeal-agar medium. The flies were incubated at 30°C, and survival was monitored daily for up to 4 days post-infection (dpi).

## Ex vivo murine ear tissue adhesion assay

An ex vivo murine ear tissue adhesion assay was performed to evaluate the adhesion capacity of *C. auris* strains. Mouse ears of equal size were trimmed, cleaned with PBS. For the adhesion assay, each *C. auris* strain was cultured overnight in 2 mL of YPD medium at 30°C with shaking at 220 rpm. Cells were harvested, washed twice with PBS, and resuspended in 10 mL of PBS and $OD_{600}$ of 0.2. Each ear was incubated in the suspension at room temperature with shaking 80 rpm for 1 or 2 h. After incubation, ears were washed three times with PBS to remove non-adherent cells, and then either incubated further on YPD plates for imaging or subject to DNA extraction for quantitative analysis. The ears were incubated in phenol:chloroform for 30 min to detach all surface-adherent fungal cells along with epidermal cells. The resulting suspension was then subjected to bead beating to extract total DNA. For quantitative analysis of adhesion, a *CNB1* gene-specific primer set was used to detect WT and mutant strains by qPCR. Fungal burden was quantified by comparing Ct values with those obtained from uninfected ear tissue controls.

## Subcutaneous murine infection model

*C. auris* wild-type and mutant strains were cultured overnight in YPD medium at 30°C with shaking. The cell concentration was adjusted to $10^8$ cells/mL, and 100 µL of the suspension in PBS was injected subcutaneously into anesthetized mice. SPF/VAF-confirmed, inbred 6-week-old female BALB/cAnNCrlOri mice (ORIENT BIO INC., South Korea) were housed for one week prior to infection. Anesthesia was induced by inhalation of isoflurane vapor (Hana Pharm. Co., Ltd., South Korea) at a flow rate of 80 cc/min. Subcutaneous injection was performed on both flanks of the mice, where the fur had been trimmed with clippers. The size of abscesses formed post-infection was quantified using ImageJ with a ruler as a reference. Colony-forming units (CFUs) were determined by surgically extracting the abscess, homogenizing the tissues, and plating the homogenates on YPD medium for colony enumeration.

## Systemic murine infection model

The cell concentration was adjusted to $10^8$ cells/mL, and 100 µL of the suspension in PBS was injected intravenously into restrained mice. Humane endpoints were defined by the onset of systemic infection symptoms, including rapid weight loss, abnormal head tilt, and circling behavior. Survival was monitored and represented as a survival curve. For drug treatment groups, drugs were administered intraperitoneally starting from the day of infection. Each drug was prepared in a 5% Kolliphor solution (polyoxyl-35 castor oil:ethanol = 1:1), and the injection volume was limited to 100 µL. The stock concentration for a 20 mg/kg (mpk) dose was 4 mg/mL, while for a 2 mpk dose, the stock concentration was 0.4 mg/mL.

## Supporting information

**S1 Table. Strains used in this study.**
(DOCX)

**S2 Table. Primers used in this study.**
(DOCX)

**S1 Fig. Phylogenetic analysis of Cna1 and Cnb1.** Phylogenetic trees for fungal orthologs for Cna1 (A) and Cnb1 (B) across various fungal species were constructed using data from the *Candida* Genome Database (http://www.candidagenome.org/).
(TIF)

**S2 Fig. Construction and verification of calcineurin gene deletion mutants and complemented strains in *C. auris*.** (A-C) Schematic representation of the homologous recombination strategies used to delete *CNA1*, *CNB1*, and both genes (left panels). Southern blot analyses confirm the successful deletion of the target genes (right panels). (D, E) Verification of the constructed complemented strains by diagnostic PCR.
(TIF)

**S3 Fig. Qualitative spot assays showing stress response of calcineurin mutants.** WT (B8441), *cna1Δ* (YSBA99), *cna1Δ::CNA1* (+*CNA1*; YSBA110), *cnb1Δ* (YSBA102), *cnb1Δ::CNB1* (+*CNB1*; YSBA111), and *cnb1Δ cna1Δ* (YSBA172) strains were spotted on YPD medium supplemented with stressors such as 50 µg/mL 5FC, 3 µg/mL FDX, 2.5 mM DIA, 0.06 mM MD, 10 mM HPX, 2.1 mM *tert*-butyl hydroperoxide (TBH), 150 mM hydroxyurea (HU), 0.03% MMS, 1.5 M NaCl, 1.5 M KCl, or 2 M sorbitol. WT and mutant strains were spotted on YP medium supplemented with 1 M NaCl, 1 M KCl, or 2 M Sorbitol. Plates were incubated for 2 or 3 days. Abbreviations: 5FC, 5-flucytosine; FDX, fludioxonil; DIA, diamide; MD, menadione; HPX, hydrogen peroxide; TBH, *tert*-butyl hydroperoxide; HU, hydroxyurea; MMS, methyl methanesulphonate; KCR, YPD + KCl; NCR, YPD + NaCl; SBR, YPD + sorbitol; KCS, YP + KCl; NCS, YP + NaCl; SBS, YP + sorbitol.
(TIF)

**S4 Fig. MIC tests of echinocandins and checkerboard assays with FK506, cyclosporin A, and echinocandins.** (A) EUCAST MIC test results for micafungin (MIF) and anidulafungin (ANF) in the WT (B8441), *cna1Δ* (YSBA99), *cnb1Δ* (YSBA102), and *cnb1Δ cna1Δ* (YSBA172) strains. (B) Checkerboard assay results showing the interaction between MIF or ANF and FK506 in the WT strain (B8441). (C) Checkerboard assay results of caspofungin (CAF), MIF, or ANF with cyclosporin A (CsA) in the WT strain (B8441).
(TIF)

**S5 Fig. Checkerboard assays with FK506, cyclosporin A, and azoles.** (A) Checkerboard assay results showing the interaction between voriconazole (VRC), posaconazole (PSC), or itraconazole (ITC) and FK506 in the WT strain (B8441). (B) Checkerboard assay results depicting the interaction of fluconazole (FLC), VRC, PSC, or ITC with cyclosporin A (CsA) in the WT strain (B8441).
(TIF)

**S6 Fig. Quantitative measurements of ergosterol contents in calcineurin deletion mutants.** (A, B) Ergosterol content was measured by spectrophotometric absorbance after extraction using the HETENE method. (A) Absorbance profiles representing basal ergosterol levels in each strain. Dashed lines indicate individual strains, with colors corresponding to the strain legend shown within the graph. (B) Ergosterol content under amphotericin B (AMB, 3 µg/mL) treatment. Non-dashed lines represent strains under drug-treated conditions, allowing direct comparison with the basal profiles in (A). (C) WT, *cna1Δ*, and *cnb1Δ* were grown to mid-log phase, lyophilized, and extracted with 25% alcoholic KOH. Sterols were isolated using *n*-heptane and quantified by measuring absorbance at 282 nm and 230 nm. Ergosterol content was calculated based on OD values normalized to dry cell weight. Statistical significance was evaluated using one-way ANOVA with Bonferroni's multiple-comparison test (NS, not significant).
(TIF)

**S7 Fig. Membrane permeability was modestly increased in the calcineurin deletion mutants.** Cells were stained with propidium iodide (PI) and analyzed by flow cytometry to assess membrane permeability following ethanol fixation and RNase A treatment. Representative dot plots are shown with PI fluorescence (610 nm, *x*-axis) and side scatter (*y*-axis). Quadrants were defined as follows: Q2 (PI+/SSC^high) and Q3 (PI+/SSC^low) represent cells with compromised membrane integrity. The percentage of PI-positive cells (Q2 + Q3) is indicated in each plot. All samples were stained with 50 µg/mL PI for 15 min at 25°C in the dark and analyzed using a CytoFLEX flow cytometer (Beckman Coulter). Data shown are representative of at least three independent experiments.
(TIF)

**S8 Fig. Construction and verification of *cna1Δ* mutants in different clades of *C. auris*.** (A-C) Diagrams illustrating the homologous recombination strategies used to delete *CNA1* in clade II B11220 (A), clade III B11221 (B), and clade IV B11245 (C) strains (left panels). Successful deletion of *CNA1* was confirmed via Southern blot analysis (right panels). (TIF)

**S9 Fig. *CNA1* expression levels in wild-type strains from different clades under basal and CFW-treated conditions.** qRT-PCR analysis of CNA1 gene in wild-type strains (B8441, B11220, B11221, and B11245). Cells were cultured with 0.5% calcofluor white (CFW) for 2 or 4 h at 30°C in a shaking incubator. Gene expression levels were normalized to *ACT1*, and fold changes were calculated relative to the basal expression level in each wild-type strain. Statistical significance was evaluated using one-way ANOVA with Bonferroni's multiple-comparison test (*, $P < 0.05$; **, $P < 0.01$; ***, $P < 0.001$; ****, $P < 0.0001$; NS, not significant). (TIF)

**S10 Fig. Identification of potential downstream transcription factors of calcineurin.** (A) Phylogenetic analysis of Crz1 and Crz2 orthologs across various fungal species. (B) Protein domain analysis of Crz1 and Crz2 in *C. auris*. (C and D) Predicted structure of zinc finger $C_2H_2$-type domain of Crz1 and Crz2, generated using AlphaFold2 (ColabFold v1.5.2). (TIF)

**S11 Fig. Construction and verification of *crz1Δ*, *crz2Δ*, and *crz1Δ crz2Δ* mutants and complemented strains in *C. auris*.** (A-C) Diagrams illustrating the homologous recombination strategies used to delete *CRZ1* (A), *CRZ2* (B), and both genes (C) in the B8441 strain (left panels). Successful gene deletion was confirmed via Southern blot analyses (right panels). (D) Verification of the *crz1Δ::CRZZ1* complemented strains by diagnostic PCR. (TIF)

**S12 Fig. Qualitative spot assays of *crz1Δ*, *crz2Δ*, and *crz1Δ crz2Δ* mutants and complemented strains in *C. auris*.** (A) Qualitative spot assays showing the stress susceptibility of WT (B8441), *crz1Δ* (YSBA105), *crz1Δ::CRZ1* (YSBA158), *crz2Δ* (YSBA143), *crz1Δ crz2Δ* (YSBA153), and *cnb1Δ cna1Δ* (YSBA172) strains. WT and mutant strains were spotted on YPD medium supplemented with stressors, including 0.5 μg/mL posaconazole (PSC), 0.8 μg/mL voriconazole (VRC), 5 μg/mL ketoconazole (KTC), 0.15 μg/mL micafungin (MIF), or amphotericin B (AMB). Plates were incubated for 2 days. (B) Phenome heat map of each mutant for various stress. Phenotype scores are color-coded based on qualitative or semi-quantitative measurements under the indicated growth conditions. Abbreviations: 30T, 30°C; 37T, 37°C; 42T, 42°C; 45T, 45°C AMB, amphotericin B; FDX, fludioxonil; 5FC, 5-flucytosine; FLC, fluconazole; PSC, posaconazole; ITC, itraconazole; VRC, voriconazole; KTC, ketoconazole; CAF, caspofungin; MIF, micafungin; ANF, anidulafungin; HPX, hydrogen peroxide; TBH, tert-butyl hydroperoxide; DIA, diamide; MD, menadione; MMS, methyl methanesulfonate; HU, hydroxyurea; TM, tunicamycin; DTT, dithiothreitol; CR, Congo red; CFW, calcofluor white; SDS, sodium dodecyl sulfate; CDS, cadmium sulfate; KCR, YPD+KCl; NCR, YPD+NaCl; SBR, YPD+sorbitol; KCS, YP+KCl; NCS, YP+NaCl; SBS, YP+sorbitol. Red and blue gradients represent phenotype reduction and enhancement, respectively, with strong, intermediate and weak phenotypes indicated by color intensity. (TIF)

**S13 Fig. Construction and verification of mCherry-tagged Crz1 strains in *C. auris*.** (A) Diagrams illustrating the homologous recombination strategies used to integrate the *CRZ1-mCherry* allele in the *crz1Δ* mutant (left panel). Diagnostic PCR confirmed the targeted integration of *CRZ1-mCherry* in the tagged strains. (B) Diagrams illustrating the homologous recombination strategies used to delete *CNA1* in the *CRZ1-mCherry* stain (left panel). Disruption of the *CNA1* in the *CRZ1-mCherry* strain was validated through Southern blot analysis. (C) Qualitative spot assays showing the phenotypes of WT (B8441), *crz1Δ* (YSBA105), *crz1Δ::CRZ1* (YSBA158), *crz1Δ::CRZ1-mCherry* (YSBA289), *cna1Δ* (YSBA99), and *crz1Δ::CRZ1-mCherry cna1Δ* (YSBA313) strains. WT and mutant strains were spotted on YPD medium supplemented with stressors, including 3 μg/mL amphotericin B (AMB), 150 μg/mL fluconazole (FLC), 1 μg/mL anidulafungin

(ANF), 0.1% SDS, 0.05% Congo red (CR), 2.5 µg/mL fludioxonil (FDX), 220 mM CdSO$_4$ (CDS), 0.1 µg/mL posaconazole (PSC), 0.8 µg/mL voriconazole (VRC), 10 µg/mL ketoconazole (KTC), 7.2 µg/mL tunicamycin (TM), or 24 mM DTT. Plates were incubated for 2 or 3 days.
(TIF)

**S14 Fig. Construction and verification of *cna1Δ mkc1Δ* in *C. auris*.** (A) Diagrams illustrating the homologous recombination strategies used to delete *MKC1* in the *cna1Δ* strain (left panels). Successful gene deletion was confirmed via Southern blot analysis (right panels). (B) qRT-PCR analysis of chitin synthase genes (*CHS1-CHS7*) and chitin deacetylase gene (*CDA2*) in WT, *cna1Δ*, *mkc1Δ*, and *cna1Δ mkc1Δ* strains. Cells were cultured for 24 h with 5 µg/mL CAF at 30°C in a shaking incubator. Gene expression levels were normalized to *ACT1*, and fold changes were calculated relative to the basal expression level in WT. Statistical significance was evaluated using one-way ANOVA with Bonferroni's multiple-comparison test (*, $P<0.05$; **, $P<0.01$; ***, $P<0.001$; ****, $P<0.0001$; NS, not significant).
(TIF)

**S15 Fig. Calcineurin is dispensable for morphogenesis, biofilm formation, SAP activity, and ploidy switching in *C. auris*.** (A) Hydroxyurea (HU)-mediated filamentous growth. Indicated WT and mutant cells were cultured overnight in YPD medium at 30°C and then subcultured into fresh YPD medium to an OD$_{600}$ of 0.8. The cultures were treated with 100 mM HU in YPD broth and incubated for 24 h at 30°C. The cells were fixed using 10% formalin and stained with Hoechst solution. Representative microscopy images of each strain are shown. (B) Biofilm formation assay. Biofilm formation by WT and mutant strains was assessed using crystal violet staining. The absorbance of the destaining solution for each strain was measured at 595 nm. The *bcy1Δ* mutant was used as a positive control. (C) Secreted aspartyl protease (SAP) activity assay. WT and mutant strains were cultured overnight, washed twice with dH$_2$O, resuspended in dH$_2$O, spotted (3 µL) onto solid YCB-BSA medium. The plates were incubated for 3 days at 37°C, and the halo diameter was measured to determine SAP activity. Experiments were biologically replicated three times. Statistical significance was assessed using one-way ANOVA with Bonferroni's multiple-comparison test (*, $P<0.05$; **, $P<0.01$; ***, $P<0.001$; ****, $P<0.0001$; NS, not significant). The *tpk1Δ tpk2Δ* and *sapa3Δ* mutants were used as negative controls. (D) Ploidy switching assessment. Approximately 100 cells from WT and mutant strains were plated on YPD medium containing 5 µg/mL phloxine B, incubated at 25°C for 14 days, and photographed. Cells isolated from phloxine B-containing plates were cultured in liquid YPD medium at 30°C for 48 h, fixed, photographed, and analyzed by FACS.
(TIF)

## Author contributions

**Conceptualization:** Yong-Sun Bahn.

**Data curation:** Hyunjin Cha, Doyeon Won, Seun Kang, Eui-Seong Kim, Kyung-Ah Lee, Won-Jae Lee, Yong-Sun Bahn.

**Formal analysis:** Hyunjin Cha, Doyeon Won, Seun Kang, Eui-Seong Kim, Kyung-Ah Lee, Won-Jae Lee, Kyung-Tae Lee, Yong-Sun Bahn.

**Funding acquisition:** Won-Jae Lee, Kyung-Tae Lee, Yong-Sun Bahn.

**Investigation:** Hyunjin Cha, Doyeon Won, Seun Kang, Eui-Seong Kim, Kyung-Ah Lee, Kyung-Tae Lee.

**Methodology:** Hyunjin Cha, Doyeon Won, Seun Kang, Eui-Seong Kim, Kyung-Ah Lee, Won-Jae Lee, Kyung-Tae Lee, Yong-Sun Bahn.

**Project administration:** Yong-Sun Bahn.

**Resources:** Won-Jae Lee, Kyung-Tae Lee, Yong-Sun Bahn.

**Supervision:** Won-Jae Lee, Kyung-Tae Lee, Yong-Sun Bahn.

**Validation:** Hyunjin Cha, Kyung-Tae Lee, Yong-Sun Bahn.

**Visualization:** Hyunjin Cha.

**Writing – original draft:** Hyunjin Cha, Yong-Sun Bahn.

**Writing – review & editing:** Won-Jae Lee, Kyung-Tae Lee, Yong-Sun Bahn.

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
