## [Decision Letter · Decision Letter 0]

PPATHOGENS-D-25-00022

The calcineurin pathway regulates extreme thermotolerance, cell membrane and wall integrity, antifungal resistance, and virulence in Candida auris

PLOS Pathogens

Dear Dr. Bahn,

Thank you for submitting your manuscript to PLOS Pathogens. After careful consideration, we feel that it has merit but does not fully meet PLOS Pathogens's publication criteria as it currently stands. Therefore, we invite you to submit a revised version of the manuscript that addresses the points raised during the review process.

Please submit your revised manuscript within 60 days Apr 13 2025 11:59PM. If you will need more time than this to complete your revisions, please reply to this message or contact the journal office at plospathogens@plos.org. Please include the following items when submitting your revised manuscript:

We look forward to receiving your revised manuscript.

Kind regards,

Chaoyang Xue, Ph.D.

Academic Editor

PLOS Pathogens

Alex Andrianopoulos

Section Editor

PLOS Pathogens

 Sumita Bhaduri-McIntosh

Editor-in-Chief

PLOS Pathogens

orcid.org/0000-0003-2946-9497

Michael Malim

Editor-in-Chief

PLOS Pathogens

orcid.org/0000-0002-7699-2064

**Additional Editor Comments :**

While we all appreciate the novelty of the study and detailed description of results and phenotypic observations, reviewers raised a number of important concerns that are centered on insufficient mechanistic depth of some of the interesting observations, e.g, how the calcineurin pathway regulates cell wall integrity and why it regulates the azole and AmB resistance so differently. Therefore, a focus of the revision should be to enhancing the mechanistic insights by providing additional supporting data and/or discussing working hypothesis for the observations. Several experiments recommended by reviewers may help enhance the mechanistic insights.

**Journal Requirements:**

1) We do not publish any copyright or trademark symbols that usually accompany proprietary names, eg ©,  ®, or TM  (e.g. next to drug or reagent names). Therefore please remove all instances of trademark/copyright symbols throughout the text, including:

- TM on page: 25.

3) Some material included in your submission may be copyrighted. According to PLOSu2019s copyright policy, authors who use figures or other material (e.g., graphics, clipart, maps) from another author or copyright holder must demonstrate or obtain permission to publish this material under the Creative Commons Attribution 4.0 International (CC BY 4.0) License used by PLOS journals. Please closely review the details of PLOSu2019s copyright requirements here: PLOS Licenses and Copyright. If you need to request permissions from a copyright holder, you may use PLOS's Copyright Content Permission form.

Potential Copyright Issues:

i) Please confirm (a) that you are the photographer of 3D, and 6D, or (b) provide written permission from the photographer to publish the photo(s) under our CC BY 4.0 license.

ii) Figures 6A, 6C, and 7. Please confirm whether you drew the images / clip-art within the figure panels by hand. If you did not draw the images, please provide (a) a link to the source of the images or icons and their license / terms of use; or (b) written permission from the copyright holder to publish the images or icons under our CC BY 4.0 license. Alternatively, you may replace the images with open source alternatives. See these open source resources you may use to replace images / clip-art:

**Reviewers' Comments:**

Reviewer's Responses to Questions

**Part I - Summary**

Reviewer #1: This study investigated the role of the calcineurin pathway in the fungal pathogen Candida auris. The CNA1 and CNB1 genes encode the catalytic and regulatory subunits of calcineurin, respectively. The researchers constructed mutants deleted for either gene or both genes. They also generated a mutant deleted for the CRZ1 gene, which encodes a transcription factor previously shown to be activated by calcineurin-mediated dephosphorylation in other organisms. Comprehensive phenotypic analyses of these mutants were conducted to assess their responses to high temperature, various antifungal drugs, and chemicals that damage the cell membrane and cell wall. Additionally, their virulence was evaluated using a fruit fly infection model as well as mouse skin and systemic infection models. The authors conclude that the calcineurin pathway employs distinct regulatory mechanisms to perform divergent roles in regulating cell wall and membrane integrity, antifungal drug resistance, and virulence in C. auris.

This is the first comprehensive investigation of the roles of the calcineurin pathway in C. auris. Relevant mutants were constructed and examined under diverse conditions, including high temperature, all three major classes of antifungal compounds, cell wall toxins, membrane-disrupting agents, and DNA-damaging agents, and assessed for virulence using three infection models. In general, experiments were well executed and controlled. The manuscript was well written with proper conclusions.

One weakness of this work is its lack of mechanistic depth in addressing some very interesting observations. For example, the high resistance of the calcineurin mutants to amphotericin B versus their susceptibility to azoles, the distinct roles of calcineurin and its downstream transcription factor CRZ1, and the likely existence of additional regulatory targets of calcineurin remain insufficiently explored. Nevertheless, these findings open exciting opportunities for future research.

Reviewer #2: This manuscript presents a carefully conducted, thorough characterization of the roles of calcineurin and the calcineurin-dependent Crz1 transcription factor in the emerging pathogenic yeast Candida auris. Using knock-out mutants for C. auris CNA1 and CNB1 encoding the two subunits of calcineurin and CRZ1, the authors have assessed the role of these genes in sensitivity to a variety of stresses including antifungals, in the expression of virulence traits and in pathogenicity using different animal models of C. auris infection.

Notable results include the demonstration that calcineurin is necessary for survival at temperatures above 42°C; for survival upon cell membrane and cell wall stress; for tolerance to azoles and echinocandins and (surprisingly) sensitivity to amphotericin B; Crz1 acts downstream of calcineurin and yet not all phenotypes associated to calcineurin loss are shared with the Crz1 knock-out mutant suggesting complex interplay between these two components of the calcium signaling pathway and the probable involvement of additional downstream effector(s) of calcineurin in this pathway; the calcineurin pathway is dispensable for important virulence traits of C. auris such a biofilm formation, morphogenesis, protease production and ploidiy shifts; and the calcineurin pathway plays contrasting roles in C. auris pathogenicity.

Many of these results are not unexpected given the knowledge that has been gained on the roles of calcineurin and Crz1 in other pathogenic fungal species such as C. albicans, C. glabrata and Cryptococcus neoformans. Yet, presentation of these results is important as there are variations across fungal species in the roles of calcineurin and Crz1. It is noteworthy that, here, the authors show that there are even variations across the different genetic cluster that structure the C. auris population. Data presented in this manuscript pave the way for future studie aimed in particular at refining our understanding of the calcium signaling pathway in C. auris and identifying additional components that explain the phenotypic differences between the calcineurin and crz1 mutants.

Reviewer #3: The manuscript by Cha et al presents data on the role of calcineurin pathway in antifungal drug resistance and viruluence of Candida auris. C. auris is a multi drug resistant human fungal pathogen that presents a serious threat to global human health. C. auris also colonizes human skin which is considered a risk factor for patient to patient transmission, especially in health care settings. Thus, the focus of the present study to identify and characterize fungal factors regulating antifungal drug resistance and skin colonization mechanisms is quite significant. The rationale for this study was built from previous published studies implicating calcineurin pathway in stress response, drug resistance and virulence in other Candida species. The manuscript is very well written. The methodology is sound and in most cases the results justify the conclusions. Overall, the findings of this work are very interesting, and may help better understand the contributions of calcineurin pathway in drug resistance and fungal pathogenesis of C. auris.

**Part II – Major Issues: Key Experiments Required for Acceptance**

Reviewer #1: The observed significant upregulation of CHS genes in the cna1 and cnb1 mutants, but not in the wild type, in response to caspofungin is intriguing. It is known that disruption of glucan synthesis activates CHS expression via the PKC-MAPK cell wall integrity pathway as a compensatory mechanism. The current findings suggest that calcineurin may play a previously unknown role in negatively regulating this pathway. It would be informative if the authors could block the PKC-MAPK pathway in the cna1 mutant to determine whether caspofungin treatment still activates CHS expression.

Reviewer #2: 1) The authors make the interesting observation that inactivation of CNA1 or CNB1 or simultaneous inactivation of the two genes leads to increased tolerance to amphotericin B (l. 180-182). They suggest that this may be explained by the role of calcineurin in regulating the ergosterol biosynthesis pathway. In order to support this hypothesis, they provide evidence that azoles trigger increased expression of ERG genes in a calcineurin-depedent manner (Fig. 2E). While this may explain the increased sensitivity of the calcineurin mutants to azoles, it does not necessarily support that the proposal that calcineurin-dependent regulation of ergosterol biosynthesis is responsible for increased amB tolerance of the calcineurin mutants. Notably, loss of calcineurin does not seem to impair ERG genes expression in the absence of azoles. To support their hypothesis, the authors should quantify ergosterol in the membranes of calcineurin mutants and possibly evidence that increasing ergosterol in the calcineurin mutants restores sensitivity to amB.

2) Similarly, it is unclear why the authors think that the increased sensitivity of the calcineurin mutants to echinocandins can be explained by the lack of change in FKS1 upregulation upon echinocandin addition and the increased upregulation of FKS2 in the calcineurin mutants. If anything, this should contribute to increased tolerance.

3) The observation that the CNA1 mutant behaves differently according to the clade background is interesting. Given that the Cna1 proteins from clade 1, 2 and 3 appear identical (Fig. 3A), it is likely that phenotypic differences between the KO mutants in these three clades is the result of different levels of expression of CNA1 or variations in the calcium signaling pathway. While exploring the second hypothesis is clearly beyond the scope of this study, providing data of CNA1 expression levels in the 4 clades would be interesting.

Reviewer #3: no major issues with the manuscript

**Part III – Minor Issues: Editorial and Data Presentation Modifications**

Reviewer #1: The cna1 and cnb1 mutants’ significantly susceptibility to azoles and high resistance to amphotericin B is both interesting and puzzling. Many azole-resistant fungal strains exhibit cross-resistance to amphotericin B as both classes of drugs target ergosterol. This deserves more discussion. Does the crz1 mutant also show heightened resistance to amphotericin B?

Lines 67-69. The description of calcineurin activation is not clear. I thought that the simultaneous binding of Ca/calmodulin to Cna1 and calcium to Cnb1 leads to calcineurin activation, as you correctly show in Fig 7.

Lines 74-75, in yeast

Lines 174. The observed fungicidal activity is a combined effect of FK506 or cyclosporin A with echinocandins. Thus, it is not correct to say that ‘both FK506 and cyclosporin A demonstrated fungicidal antifungal activity.’ Also, in the description from line 173 to 179, echinocandin concentrations used should be mentioned.

Line 190, The result of ERG6 expression shown in Fig 2E should be described here. It is not mentioned anywhere in the text.

Lines 197-199, How does the significantly increased expression of FKS2 in the cna1 mutant explain its high susceptibility to echinocandins? Shouldn’t the increased expression of a drug target increase resistance to the drug? Need more discussion.

Lines 265-266, The subtitle is confusing. Did you mean to say ‘Crz1 and calcineurin play distinct roles in resistance to cell wall-damaging stressors and echinocandins in C. auris.’?

Reviewer #2: - l. 50: the figure drawn from reference 1 has been debated, maybe replace “over” by “an estimated” or provide a range from drawn from different studies.

- l. 53-55: “often”, “high mortality rates”: it would be preferable to provide figures, ranges for these statements

- l. 68: revise “altering its conformation of Cna1”

- l. 102: provide a reference for CGD

- l. 111-112: How robust is the Alphafold3 prediction for these two proteins and the complex they form relative to existing 3D structures? Add a reference for Alphafold3.

- l. 121: add references

- l; 128: here or in the discussion, the authors might want to discuss the relevance of high temperatures (>42°C) in C. auris lifestyle…

- l. 148: it would probably be more appropriate to use tolerance instead of resistance as the ability of C. auris to grow at the studied concentrations of antifungals is an intrinsic property of the species rather than a genetically-acquired change in phenotype.

- l. 164-165: Fig. 2B is a representation of the EUCAST test for MIC. It does not provide evidence that azoles become fungicidal in the absence of calcineurin. To demonstrate this, the authors have added one step that is described in the methods section. This should be indicated in order to support the statement (the same comment applies to the statement l. 178).

- l. 177: why aren’t the actual FIC values given in the text?

- l. 227: evolutionary

- l. 229: it is surprising that the authors do not report the sensitivity of the crz1 mutant to extreme temperatures

- l. 271: why “still”

- l. 332: should read bcy1

- l. 343-345: the interest of using a subcutaneous model of infection is unclear given the physiopathology of C. auris. one would argue that skin colonization rather than subcutaneous infection should be preferentially studied.

Reviewer #3: The authors report very interesting data with azole, echinocandin and polyene susceptibity / resistance. As noted by the authors, it is quite intriguing opposite phenotypes of drug susceptibility of cna and cnb mutants with azoles and amphotericin B. The authors hypothesize that these changes may be due to increased membrane permeability of cna or cnb mutants. Thus, it may be of interest to perform either rhodamine 6g or fluorescein diacetate (FDA) uptake assay with these mutants to demonstrate membrane permeability changes in these strains.

PLOS authors have the option to publish the peer review history of their article (what does this mean? ). If published, this will include your full peer review and any attached files.

**Do you want your identity to be public for this peer review?** For information about this choice, including consent withdrawal, please see our Privacy Policy .

Reviewer #1: No

Reviewer #2: No

Reviewer #3: No

**Figure resubmission:**
---

## [Decision Letter · Decision Letter 1]

Dear Dr. Bahn,

We are pleased to inform you that your manuscript 'The calcineurin pathway regulates extreme thermotolerance, cell membrane and wall integrity, antifungal resistance, and virulence in Candida auris' has been provisionally accepted for publication in PLOS Pathogens.

Best regards,

Chaoyang Xue, Ph.D.

Academic Editor

PLOS Pathogens

Alex Andrianopoulos

Section Editor

PLOS Pathogens

Sumita Bhaduri-McIntosh

Editor-in-Chief

PLOS Pathogens

orcid.org/0000-0003-2946-9497

Michael Malim

Editor-in-Chief

PLOS Pathogens

orcid.org/0000-0002-7699-2064

Reviewer Comments (if any, and for reference):

Reviewer's Responses to Questions

**Part I - Summary**

Reviewer #1: This is a revision, and the authors have satisfactorily addressed all the issues I raised previously.

Reviewer #3: (No Response)

**Part II – Major Issues: Key Experiments Required for Acceptance**

Reviewer #1: NO further experiments are needed.

Reviewer #3: (No Response)

**Part III – Minor Issues: Editorial and Data Presentation Modifications**

Reviewer #1: No minor issues.

Reviewer #3: (No Response)

PLOS authors have the option to publish the peer review history of their article (what does this mean? ). If published, this will include your full peer review and any attached files.

**Do you want your identity to be public for this peer review?** For information about this choice, including consent withdrawal, please see our Privacy Policy .

Reviewer #1: No

Reviewer #3: No